# Closed-loop direct control of seizure focus in a rodent model of temporal lobe epilepsy via localized electric fields applied sequentially

Wonok Kang [1,2], Chanyang Ju[2,3], Jaesoon Joo[4], Jiho Lee[2,3], Young-Min Shon[4,5] ✉ & Sung-Min Park [1,2,3,6,7,8] ✉

Direct electrical stimulation of the seizure focus can achieve the early termination of epileptic oscillations. However, direct intervention of the hippocampus, the most prevalent seizure focus in temporal lobe epilepsy is thought to be not practicable due to its large size and elongated shape. Here, in a rat model, we report a sequential narrow-field stimulation method for terminating seizures, while focusing stimulus energy at the spatially extensive hippocampal structure. The effects and regional specificity of this method were demonstrated via electrophysiological and biological responses. Our proposed modality demonstrates spatiotemporal preciseness and selectiveness for modulating the pathological target region which may have potential for further investigation as a therapeutic approach.

Epilepsy is one of the most common neurological disorders that afflicts almost 70 million people worldwide[1]. Due to its high irregularity and unpredictability, epileptic seizures often result in physical injuries, such as broken bones, and can severely disrupt a person's daily life[2]. In addition, epilepsy has a significant impact on the quality of life of those who have the condition and their families, as it is often accompanied by psychological disorders including depression and anxiety[3,4]. Epileptic seizure is thought to be caused by abnormally excessive or synchronous electrophysiological activity in the brain[5]. Accordingly, to suppress or eliminate neural circuit dysfunction, conventional therapeutic modalities, such as anti-seizure medication (ASM) or resective surgery, have been used for many decades. Despite optimal treatment with ASM, approximately 30% of patients have known to be drug-refractory[6]. Surgical resection of the epileptogenic zone can be an effective therapeutic option for cases with unitary ictal onset but is not suitable for overlying eloquent areas or regions with multiple epileptogenic zones[7,8]. Recently, deep brain stimulation (DBS), which provides control of epileptic circuits in a spatial and temporal manner,

has been introduced as an alternative and less-invasive therapeutic modality for intractable epilepsy compared with the surgical approach[9]. While multiple brain regions such as the hippocampus, the anterior nuclei of the thalamus (ANT), the centromedian thalamic nucleus (CM), and the motor cortex have been identified as legitimate sites for delivering stimulation therapy, the optimal target selection and stimulation parameters remain as the subject of extensive debate[3].

The hippocampus has been considered a promising stimulation target for epilepsy due to the intrinsic anatomical connectivity related to the generation and propagation of epileptic seizures in temporal lobe epilepsy (TLE), which is the most prevalent type of epilepsy[10]. Numerous studies have been conducted investigating the stimulus parameters and treatment efficacy of hippocampal stimulation, and the results support the promise of hippocampal stimulation as a therapeutic method for ameliorating epilepsy[10–12]. An in vivo study comparing anti-seizure effects during unilateral versus bilateral hippocampal DBS in a rat model of TLE proposed that targeting larger regions of the hippocampus may provide a higher ictal suppression

[1]School of Interdisciplinary Bioscience and Bioengineering, Pohang University of Science and Technology, Pohang 37673, Republic of Korea. [2]Medical Device Innovation Center, Pohang University of Science and Technology, Pohang 37673, Republic of Korea. [3]Department of Convergence IT Engineering, Pohang University of Science and Technology, Pohang 37673, Republic of Korea. [4]Biomedical Engineering Research Center, Samsung Medical Center, School of Medicine, Sungkyunkwan University, Seoul 06351, South Korea. [5]Department of Neurology, Samsung Medical Center, School of Medicine, Sungkyunkwan University, Seoul 06351, Republic of Korea. [6]Department of Electrical Engineering, Pohang University of Science and Technology, Pohang 37673, Republic of Korea. [7]Department of Mechanical Engineering, Pohang University of Science and Technology, Pohang 37673, Republic of Korea. [8]Institute of Convergence Science, Yonsei University, Seoul 03722, Republic of Korea. ✉e-mail: youngmin.shon@samsung.com; sungminpark@postech.ac.kr

potency[13]. In addition, several clinical reviews of DBS targeting techniques suggested that the relatively large structure of the hippocampus can have an advantage in determining electrode configuration compared with other deep brain tissue with rather small volumes, as is the case for the ANT and the CM[14,15]. Additionally, since the seizure onset zone in TLE is commonly located in the hippocampus, it is plausible that direct modulation targeting the hippocampal formation may immediately terminate or suppress epileptic networks, implying that the early termination of seizures in TLE may be theoretically possible using hippocampal stimulation[16,17]. While these scientific findings support the clinical advantage of hippocampal stimulation for epilepsy treatment, the clinical application of hippocampal stimulation is still in its infancy and entails multiple challenges for developing mandatory DBS therapy-related knowledge, such as the spatial characteristics of the stimulation target and its connected networks, and optimal stimulation parameters[18,19]. Furthermore, there remains a concern that a stimulus applied broadly using a conventional DBS method called wide-field (WF) stimulation to the large hippocampal structure may induce unsuitable or excessive effects on its adjacent structures, such as the amygdala, the entorhinal cortex, and the parahippocampal gyrus, resulting in diverse adverse effects, such as memory impairment and emotional disorders[20–22]. Thus, an innovative method that can modulate the large bilateral hippocampal formation with high fidelity, without the unintended stimulation of neighboring brain tissue, represents an imminent unmet need for enabling optimal stimulation therapy for epilepsy, as well as other drug-refractory neurological disorders.

In this study, we present a highly localized and temporally organized electrical stimulation modality called sequential narrow-field (SNF) stimulation for the rapid termination of seizures while concurrently preventing undesired tissue stimulation. To determine the therapeutic feasibility of overall hippocampal stimulation in TLE underlying the proposed method, we first investigated the anti-seizure effect with a broadly applied stimulus field at fixed rates for the entire hippocampus, confirming that 70% of electrographic seizures were terminated. Second, we investigated the phase synchrony of neuronal circuits in both hemispheres during on-demand hippocampal control using a stimulus that was applied unilaterally versus bilaterally, and found that the bilateral configuration disrupted hyper-synchronized neuronal networks better. Third, we designed and implemented a SNF stimulation method with minimized fringing electric fields that could trigger unintended neuronal responses (Fig. 1). Fourth, we verified the effectiveness of this method by comprehensively analyzing the physiological and histological responses during stimulation via extensive in vivo and in silico methods. Finally, we investigated the therapeutic effects and cognitive-behavioral responses during SNF stimulation in a survival model, and confirmed the validity of translational relevance for human TLE.

## Results

### Sequential narrow-field hippocampal stimulation
A conceptual overview of the proposed on-demand SNF pulse stimulation modality and its operating principles are shown in Fig. 1. The depth electrodes inserted into the bilateral hippocampal region are used to record local field potential (LFP) and to stimulate the entire hippocampal structure (Fig. 1a). The LFPs recorded from the left and right hippocampi are used to identify abnormal brain activity in real-time. When electrographic seizures are detected based on the specific band power of the LFP (Methods), short stimulus pulses are applied to the electrode array, and as the result, multiple localized electric fields are induced in a sequential manner to intervene in the ictal activity (Fig. 1a, left). Our hypothesis proposes that the localized gradients induced sequentially for each section extensively modulate the entire bilateral hippocampal structure as if a single high-frequency electric field is applied due to the relatively slow axonal conduction[23–26]. In this

way, the integrated gradient can intervene in the network rhythm in only the target region (Methods, Supplementary Fig. 1). The diagram with matrices presenting the neuronal synchronization level conceptually illustrates a more detailed therapeutic mechanism and the advantage of the proposed stimulation method (Fig. 1b). If initial electrographic seizure activity in TLE appears in a restricted region, without timely intervention, it progresses to other brain regions (Fig. 1c). Ictal activity, if detected in the early stage, can be terminated by proper control of the specific region related to the pathological synchronization of epileptic seizures, thus hippocampal stimulation may be an appropriate therapeutic strategy in TLE. However, a stimulus that is broadly applied at once to entirely modulate the large hippocampal structures would likely induce fringing fields spreading outside the target tissue, thus leading to unintended stimulation outside the target area (Fig. 1d). In summary, the proposed SNF stimulation method can effectively control the entire region of the hippocampus to terminate seizures with sequentially induced multiple localized fields, thereby avoiding fringing field effects that may elicit unwanted and excessive stimulation-related responses outside the region of interest (Fig. 1e).

### Anti-seizure effects of hippocampal intervention
To demonstrate the therapeutic efficacy of the proposed SNF stimulation modality, we first evaluated the therapeutic potential of the overall hippocampal modulation in TLE using a pre-programmed stimulus that repetitively intervened in the hippocampal network at fixed rates (Fig. 2). Intrahippocampal depth electrodes were bilaterally implanted to modulate the entire hippocampal region and to monitor brain activity (Fig. 2a). Following the recovery period for electrode implantation, the acute status epilepticus (SE) model was induced by intraperitoneal kainic acid (KA) injection (Fig. 2b, Methods). This KA-induced SE model reproduces the neuropathological and electroencephalographic features in patients with TLE[27]. Then, electrographic seizures were rapidly suppressed by stimulation compared with non-stimulated ones (Fig. 2c–f; representative results of LFP traces with and without intervention). Especially, stimulation succeeded in suppressing seizures not only in rapid interventions (Fig. 2c) but also in interventions for fully established epileptic rhythms (Fig. 2d), indicating strong seizure-suppressing effects of the overall hippocampal control. Approximately 70% of seizures were terminated with hippocampal stimulation that intervened in ongoing ictal activities (Fig. 2g, h; termination latency after stimulation: 3.13 (median) and 5.61 (mean) (IQR (interquartile rate) = 1.5–7.75) s; n = 204 trials from 6 animals). These results demonstrate the potential of the overall hippocampal stimulation to advance as a seizure treatment technique in TLE.

### Comparison of neuronal desynchronization by bilateral versus unilateral stimulation
Epileptic seizures in patients with TLE can originate from the unilateral or bilateral hippocampus, and similar to that, the ictal activity in the KA-induced SE model can be initiated from the hippocampal region unilaterally or bilaterally[28,29]. Thus, numerous studies on unilateral (including ipsilateral and contralateral approaches) and bilateral DBS regarding the onset zone have been conducted as part of both preclinical experiments and patient trials[10,17,30,31]. In addition, it has been reported that bilateral hippocampal control has superior efficacy in terms of suppressing seizures, even in unilateral TLE[13,32]. However, meticulous evaluation of the therapeutic effects of overall hippocampal stimulation with a bilateral configuration compared with a unilateral one has rarely been investigated. Therefore, we asked whether the unilaterally or bilaterally applied WF stimulation (i.e., electric field applied broadly) to the entire hippocampal formation in a timely manner leads to different synchronization of neuronal networks[33,34] during seizures and, consequently, results in a gap in inhibitory efficacy (Fig. 3). To test this, we implemented a closed-loop seizure

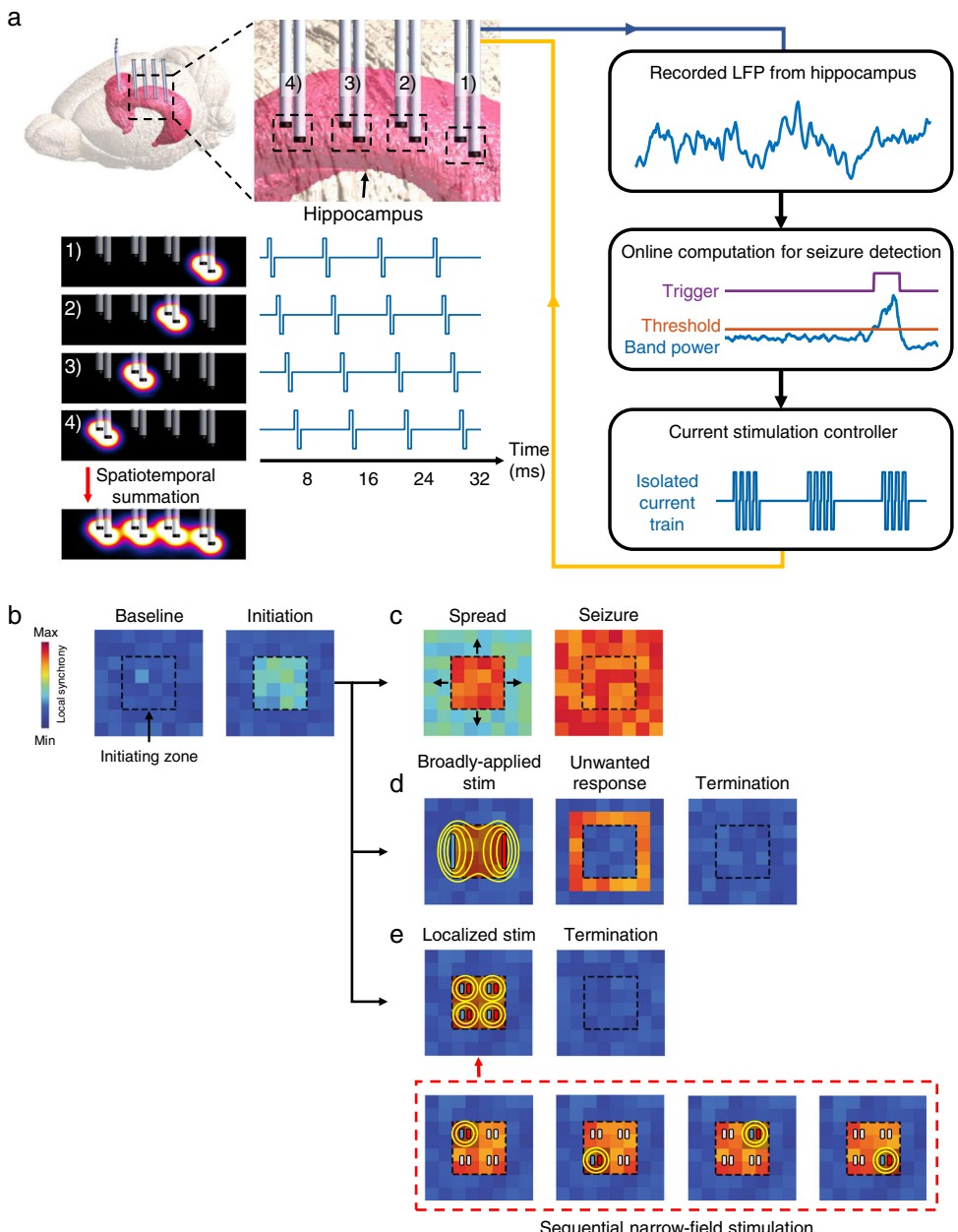

**Fig. 1 | Overview of the SNF stimulation strategy. a** Diagram of the closed-loop on-demand SNF stimulation system. Multi-channel electrode arrays are implanted in the bilateral hippocampal regions. The recorded signals from multiple locations are fed into the real-time seizure detection algorithm based on features of signal frequency and power. Once a seizure has been detected with the computed thresholds for each channel, localized stimulus pulses are applied to the overall hippocampal structure in a sequential manner. **b** Conceptual illustration of the on-demand SNF stimulation modality mechanism. **c** A generalization of seizures without intervention. **d** Unwanted neural activation by fringing field effects induced by broadly applied stimulation. **e** Seizure termination with sequentially applied localized stimulation.

control system in both unilateral and bilateral configurations and investigated the phase synchrony index between both hemispheres (Fig. 3a–e, Methods). The phase of the signals was extracted and is shown in Fig. 3b. During seizures, the phases of both hemispheres were very similar, resulting in the phase difference concentrating at 0 degrees (Fig. 3c). Then, the phase synchrony index during stimulation was investigated using the phase difference between both hemispheres (Fig. 3d, e; Supplementary Fig. 2a). The synchrony index of the baseline (before KA injection) was skewed to the left (where low values dominated), while that of the non-stimulation group was skewed to the right (where high values dominated), as shown Fig. 3d. The results from the unilateral and bilateral WF stimulation groups were located between the distribution of the baseline and the non-stimulation

group and show a significant gap in phase synchrony near the index of 0.7 (Fig. 3d), reported as paroxysmal values[34]. The synchronization index at the baseline occupied the highest ratio in the normal phase range, followed by groups for bilateral stimulation, unilateral stimulation, and non-stimulation, indicating that stable brain activity has a high proportion in the normal rhythm (Fig. 3e, left). Bilateral WF stimulation significantly suppressed the hyper-synchronized neural rhythm compared with a unilateral configuration (Fig. 3e, right). The success rate of terminating seizures by unilateral and bilateral control was analyzed every 20 min, and significant differences were found between them (Fig. 3f; two-way repeated measures ANOVA; $P = 0.006$ and Supplementary Table 1). As a result, the total seizure duration of bilateral stimulation was 18.6% less than that of unilateral one (Fig. 3g;

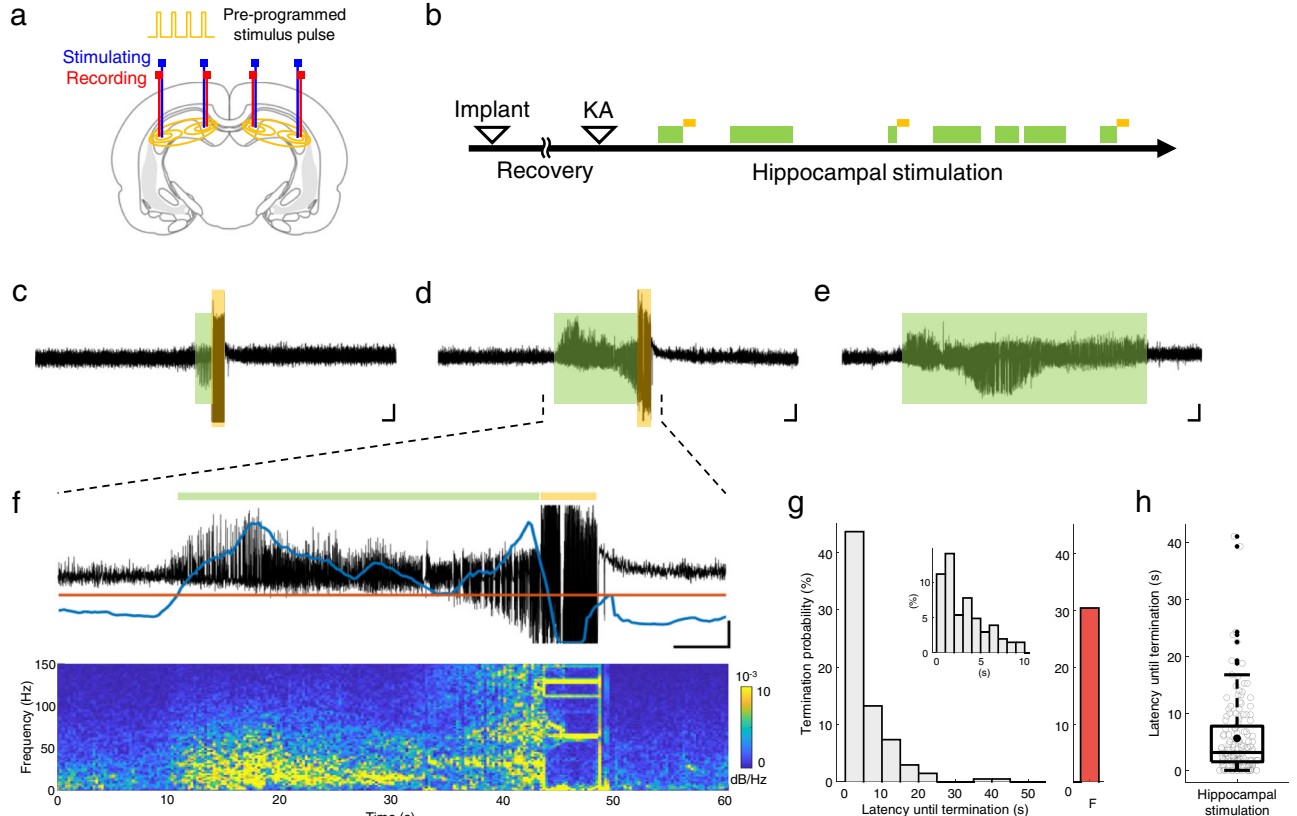

**Fig. 2 | Seizure control in a KA-induced SE model via overall hippocampal intervention. a** Schematic depicting the insertion positions of the depth electrode and the stimulation pattern. **b** Experimental timeline. **c–f** Representative recording results of electrographic seizures with hippocampal control in **c** and **d** or without control in **e** (seizures detected: green boxes; electrical stimulus: vertical orange line). **f** Excessive spontaneous spike-wave discharges detected by the algorithm (top; current index: blue line; threshold: orange line) and their corresponding power spectrum (bottom; using a 0.5 s sliding window with 50% overlap).

**g** Distribution of latencies until seizure termination (5 s bin size) by hippocampal stimulation (control success: gray; failed control: red). Inset: expanded 1 s bin size for a 0 to 10 s range. **h** Box plot showing the mean and range of termination latencies ($n = 204$ trials from 6 rats). Center bar and dot in box indicate median and mean for each, box indicates 25th and 75th percentiles, and whiskers extend to the most extreme data points without outliers (±1.5 IQR). Outliers are shown in dot signs. Scale bars in **c–f**, 5 s, 1 mV. F failed. Source data are provided as a Source Data file.

20.3% and 38.9% for bilateral and unilateral WF stimulation, respectively; two-tailed Mann–Whitney $U$-test; $P = 0.0226$ for comparison between the two conditions; and $P = 0.9362$ for all in-group comparisons between left and right hemisphere), and it resulted in a significant decrease in band power during bilateral WF stimulation compared to unilateral stimulation (Supplementary Fig. 2c). Taken together, these findings demonstrate that bilateral hippocampal stimulation can effectively desynchronize the neuronal networks in paroxysmal periods compared with the unilateral configuration, resulting in a significant difference in seizure-inhibitory effect.

## Validation of the SNF stimulation method

The therapeutic efficacy of overall hippocampal stimulation in TLE was demonstrated by closed-loop WF stimulation. However, as previously noted, due to its fringing field effects, which would likely result in undesired gradient distribution in regions adjacent to the target area, WF stimulation can lead to unintended neuronal responses. In this context, we asked whether the proposed SNF stimulation method could induce the effectiveness of the overall hippocampal stimulation for suppressing the seizure activity while avoiding unwanted stimulation. If possible, we can develop a superior hippocampal stimulation than conventional WF stimulation in terms of effectiveness and safety. To determine this, we first measured the intracerebral electric fields induced by WF and SNF stimulation and quantitatively compared the difference in induced electric fields outside the hippocampal structure (Fig. 4a–h). Depth electrode arrays for generating stimulation and

recording the induced electric potential were implanted into the hippocampus and across the entire brain, respectively (Fig. 4a). Stimulation currents were individually applied to each five electrode configurations (Fig. 4a, bottom right), and the intracerebral potential was recorded for calculating the induced voltage gradients between electrodes (Fig. 4b; Supplementary Fig. 3). The electric field distribution due to the broadly applied current spread significantly farther than the induced field due to the narrowly applied current (Fig. 4b, c and Supplementary Table 1). In order to see the difference in activated volume due to stimulation pulses, the proportions of the area where the off-target spreading field exceeds 10 mV/mm (widely accepted as necessary to affect the associated network rhythms[35]) and 50 mV/mm (reported to be required to suppress seizures with pulsed stimulation[36,37]) were compared for WF and SNF conditions (Fig. 4d, e). In both comparisons, the results showed significant differences between WF and SNF stimulation, clearly implying the superiority of SNF stimulation in preventing undesired activation or inhibition of off-target tissue over WF stimulation. The linear spreading characteristics were further investigated by considering the direction of the main field vectors (Fig. 4f; magnified view on the red dotted box in Fig. 4a), and the results showed significant differences between WF and SNF conditions, like those two-dimensionally investigated in the whole brain (Fig. 4g, h). These findings were further supported by estimating the volume of activated area in adjacent brain regions including the cortex, thalamus, midbrain, and striatum via in silico method (Supplementary Fig. 4) and investigating brain regions activated during

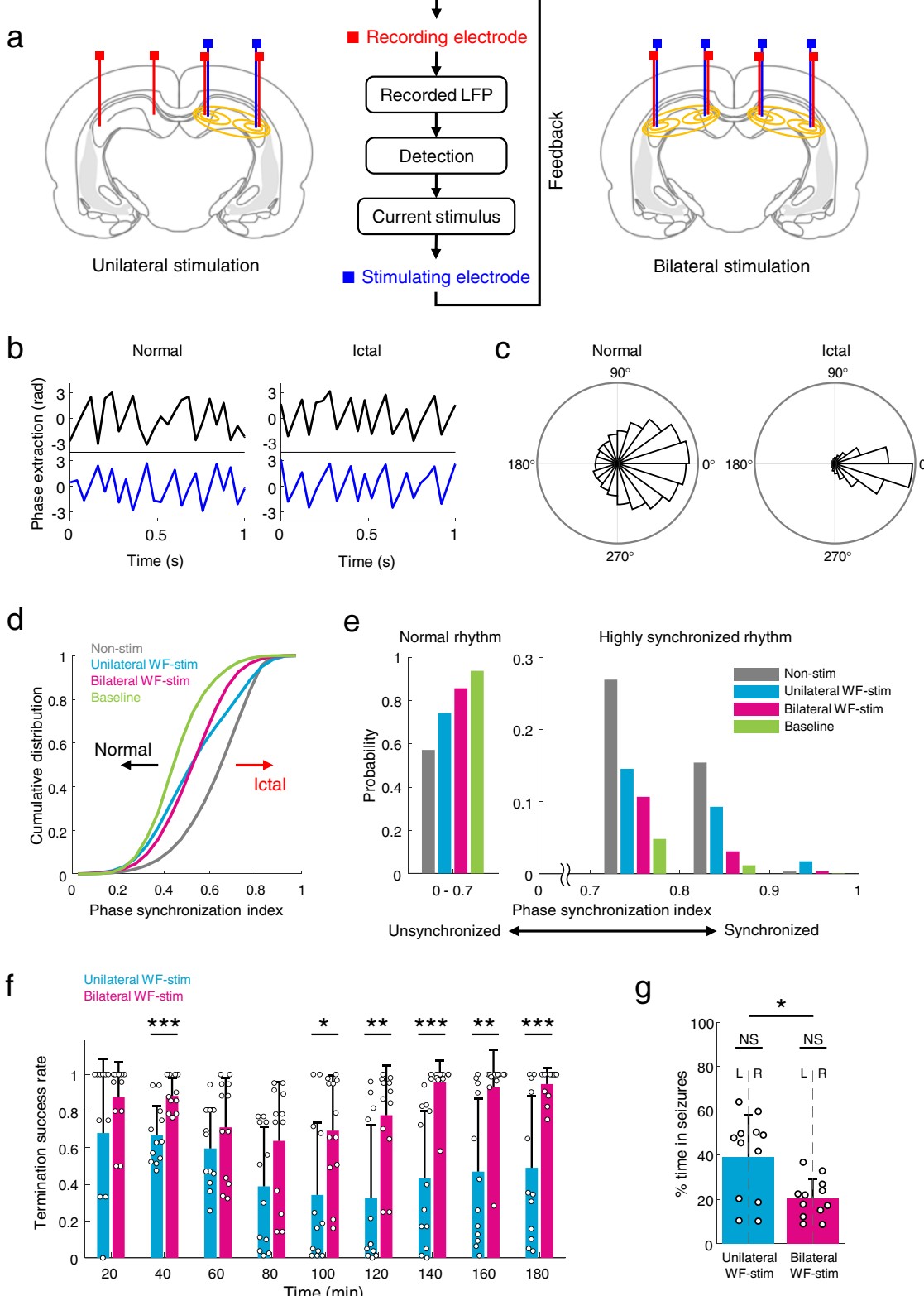

stimulation using the c-Fos as a marker of neural activity (Supplementary Fig. 5). However, since the neuronal membrane could temporally integrate multiple fields with similar vector directions due to their intrinsic properties[38,39], we cannot easily conclude that SNF stimulation does not induce unwanted stimulation compared with WF stimulation. Therefore, we compared the neuronal responses in the primary motor cortex (M1) elicited by the SNF and WF stimulation methods using a computational model that could reproduce the neuronal characteristics, integrating multiple electrical gradients applied in a very short time (Fig. 4i, j)[38,40,41]. As a result, the normalized threshold for neural responses in the M1 by hippocampal SNF stimulation was significantly greater compared with the WF stimulation, ranging from 2 to 9 times for each position (Fig. 4k; 1.00 and 0.30 for SNF and WF stimulation, respectively; two-tailed paired $t$-test; $P < 0.001$; $n = 18$ recording points for each group). In addition, we investigated motor responses during stimulation with varying stimulus

**Fig. 3 | Closed-loop unilateral and bilateral control of temporal lobe seizures. a** Schematic of electrode configurations for unilateral and bilateral stimulation and the feedback stimulation approach for intervention in a timely manner. **b** Examples of phase extraction from normal and ictal periods in both hippocampal regions (signals from the left and right hippocampus are black and blue lines, respectively). **c** The phase differences on the unit circle, where can be presented as a more concentrated angle distribution during the seizure state, resulting in a high synchrony index. **d** Cumulative distribution of neuronal synchronization between both hippocampi for the non-stimulation, unilateral stimulation, and bilateral stimulation groups, as well as the baseline. **e** Probability density of the synchrony index in the normal (left) and hyper-synchronized (right) range. **f** The success rate of seizure suppression with unilateral (blue) or bilateral (magenta) interventions over segmented time ($P = 0.006$, $n = 6$ rats per group, two-way repeated measures ANOVA with Bonferroni correction). **g** Percentage of time in seizure episodes during unilateral and bilateral control ($P = 0.0226$ for group comparison, $n = 12$ records per group, two-tailed Mann–Whitney $U$-test). *$P < 0.05$; **$P < 0.01$; ***$P < 0.001$; NS, not significant. Bar graphs depict data as mean ± SD. Non-stim non-stimulation, WF-stim wide-field stimulation, L Left hemisphere, R Right hemisphere. For detailed statistical information, see Supplementary Table 1. Source data are provided as a Source Data file.

intensity in freely moving rats. Similar to the numerical simulation results, SNF stimulation required an ~4.6 times larger stimulus current to evoke abnormal motor responses in comparison to WF stimulation (Fig. 4l; 658 and 142 µA for SNF and WF stimulation, respectively; two-tailed Mann–Whitney $U$-test; $P = 0.0048$; $n = 6$ animals for each group). Here with these in vivo and in silico results, we established the safety threshold of stimulus intensity representing a maximum current level that does not elicit abnormal motor and sensory responses for fair comparisons of anti-seizure effects during stimulation within the acceptable range of clinical translation (650 and 140 µA for SNF and WF stimulation, respectively). Then, we comprehensively analyzed the electrophysiological activity in the brain during acute SE for the non-stimulation, WF stimulation, and SNF stimulation groups within these safety thresholds (Fig. 5), and we termed WF stimulation at its safety level as low-intensity WF (LIWF) stimulation. The custom-made electrode arrays for SNF stimulation were bilaterally implanted into the hippocampus, in a manner as similar as possible to the electrode configuration of WF stimulation (Fig. 5a, Methods). The power of alpha (8–13 Hz), beta (13–30 Hz), and gamma (30–80 Hz) frequency activity was significantly reduced during LIWF and SNF stimulation compared with non-stimulation except for the alpha power for the LIWF condition, and furthermore, SNF stimulation further suppressed epileptic rhythms over LIWF stimulation, demonstrating that the proposed SNF modality could, at its safety level, significantly suppress excessive neuronal rhythms compared to LIWF stimulation in both the hippocampus and the cortex (Fig. 5c, e and Supplementary Table 1; Kruskal–Wallis test; $P = 0.019$, <0.001, and 0.0016 for alpha, beta, and gamma power in the hippocampus, respectively, $P = 0.01$ for alpha and $P < 0.001$ for beta and gamma power comparisons in the cortex; $n = 12$ in 6 animals for each group). Thus, the success rate of seizure termination using LIWF and SNF stimulation types significantly differed (Fig. 5f; 0.57 and 0.82 for LIWF and SNF stimulation, respectively; two-tailed Mann–Whitney $U$-test; $P = 0.0202$; n = 6 animals for each group). Similarly, the total seizure duration was remarkably reduced by SNF stimulation compared with the LIWF stimulation and non-stimulation groups (Fig. 5g; 81.5%, 55.1%, and 27.7% for the non-stimulation, LIWF stimulation, and SNF stimulation groups, respectively; Kruskal–Wallis test; $P < 0.001$; $\chi^2(2) = 15.16$; $n = 6$ animals for each group). In addition, the seizure durations during LIWF and SNF stimulation were further analyzed by segmenting them into 20 min intervals, presenting significantly different trends for most periods (Fig. 5h and Supplementary Table 1).

Experimental data from non-stimulated and SNF-stimulated groups were further analyzed to investigate the seizure-suppressing effects including early termination of hippocampal seizures and termination of fully propagated seizures, and the results showed that SNF stimulation suppressed epileptic rhythms not only in hippocampal onset but also in diffuse conditions (Supplementary Fig. 6). In addition, the anti-seizure effects during unilateral versus bilateral SNF stimulation were investigated, and the results demonstrated the superior effects of the bilateral configuration on spectral density, phase synchrony, and seizure duration during control (Supplementary Fig. 7). Overall, combined in vivo and in silico

experiments prove that SNF stimulation can effectively modulate the hippocampal region to suppress epileptic seizures while preventing unwanted stimulation.

## Biological analysis for the seizure-suppression effects of SNF stimulation

To characterize the neuromodulatory effects of hippocampal SNF stimulation on its biological consequences, our acute experimental animal brains were immunohistochemically examined following stimulation. First, to evaluate the KA-induced neuronal hyperactivation and its inhibition, immediate early gene c-Fos was investigated. In the non-stimulation group, the strong expression of c-Fos in rodent brain regions including the CA1, CA3, dentate gyrus (DG), and cortex was observed 3 h after KA injection (Fig. 6a, top), which corresponded to hyperactivity in neurons during SE as previously reported[42,43], and in the LIWF condition, no significant reductions in hyperactivated neuronal cells compared to the non-stimulated rats were observed except in the cortex (Fig. 6a, middle). However, in the SNF stimulation group, c-Fos positive neurons were expressed in significantly lower amounts than in the non-stimulation and LIWF stimulation groups throughout the entire brain, except in the DG (Fig. 6a, bottom, and b; one-way ANOVA; $P = < 0.001$, 0.0014, 0.32, and < 0.001 for the CA1, CA3, DG, and cortex regions, respectively). Second, to identify changes in neuronal inhibitory synaptic activity following stimulation, the molecular marker GAD65 was also used to analyze expression level changes in brain regions. No significant differences were found in the number of GAD65 positive cells in the brains of non-stimulated, LIWF-stimulated, and SNF-stimulated rats (Fig. 6c, d; one-way ANOVA; $P = 0.3426$, 0.2648, 0.1138, and 0.687 for the CA1, CA3, DG, and cortex regions, respectively).

We further verified the expression of Nissl bodies and ionized calcium-binding adapter molecule 1 (Iba1) to evaluate the anti-seizure effects of SNF stimulation at the cellular level by quantifying neuronal death and microglial activation in the acute phase of epilepsy (Fig. 7). The results of Nissl staining indicated no noticeable loss of pyramidal cells in the CA1, DG, and cortex regions in all conditions of the non-stimulation, LIWF stimulation, and SNF stimulation (Fig. 7a, b; one-way ANOVA; $P = 0.7833$, 0.253, and 0.7673 for the CA1, DG, and cortex regions each). While evident neuronal damage in the CA3 region, manifested by densely stained pyknotic cells and compared to a typical response during the acute phase of epilepsy as previously reported[44], was observed in both non-stimulated and LIWF-stimulated conditions, a large number of non-damaged Nissl positive cells observed in the SNF condition demonstrated that SNF stimulation prevented neuronal loss compared to other conditions (Fig. 7a, b; one-way ANOVA; $P = 0.0392$). The Iba1 is widely used as a marker of microglia, and it has been reported that excessive microglial activation may contribute to pathophysiological changes such as inflammation in the brain, a major feature of epilepsy[45,46]. In the hippocampal region and cortex, the number of Iba1 positive cells in rats that received SNF stimulation was significantly lower compared to the non-stimulation and LIWF stimulation groups (Fig. 7c, d; one-way ANOVA; $P < 0.001$ for all comparisons).

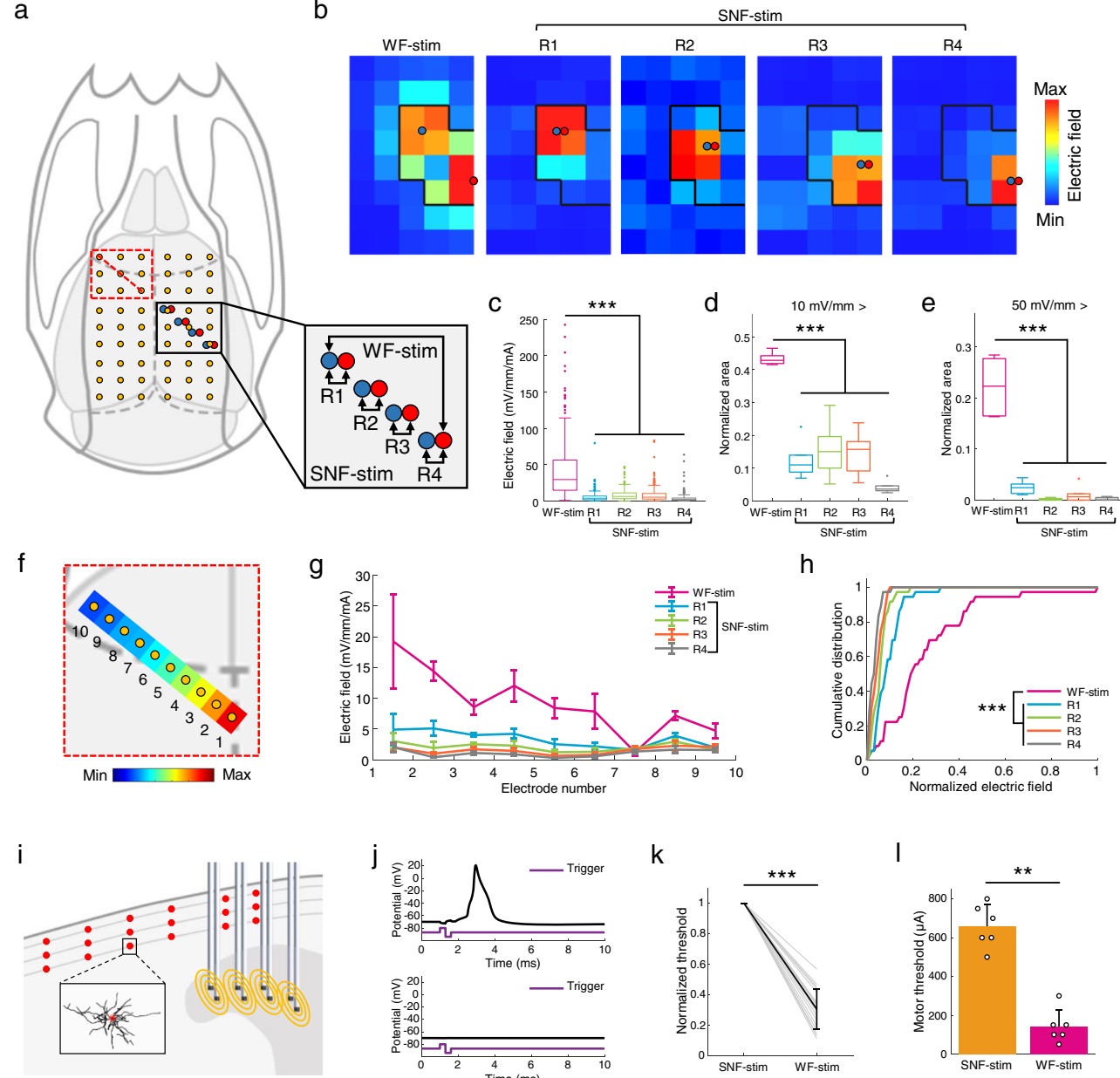

**Fig. 4 | Comparison of fringing field effects during stimulation in vivo and in silico. a** In vivo experimental setup for measuring the induced intracerebral electric fields (left). Electrode configurations for stimulation (bottom right). **b** Examples of calculated electric fields with recorded potentials. Black lines indicate hippocampal boundaries. **c**–**e** Measurements of the field distribution throughout the whole brain induced by hippocampal stimulation. **c** Normalized voltage gradients for each configuration ($P < 0.001$, $n = 180$ normalized intensities in 6 rats, one-way ANOVA with Bonferroni correction). **d, e** Comparisons of the off-target spreading field that exceeds 10 mV/mm (**d** $P < 0.001$, $n = 6$ rats, one-way ANOVA with Bonferroni correction) and 50 mV/mm (**e** $P < 0.001$, $n = 6$ rats, one-way ANOVA with Bonferroni correction). **f** Electrode array in the contralateral hemisphere to investigate the linear properties of fringing gradients, and example recorded potential. **g**, Normalized electric fields for each position ($n = 6$ rats). **h** Corresponding cumulative distributions of induced fields (P-values in Supplementary Table 1,

$n = 36$ normalized fields in four rats, Kolmogorov–Smirnov test). **i** Simulation setup for estimating the neural activation in cortical regions during hippocampal stimulation. **j** Examples of stimulus-induced action potential (top) and no response (bottom). **k** Computed thresholds for firing action potentials for each position ($P < 0.001$, $n = 18$ samples per group, two-tailed paired t-test). The full data set is superimposed in gray. The thresholds were normalized to the result of the SNF stimulation for each case. **l** In vivo measurements of stimulus-evoked abnormal motor response ($P = 0.0048$, $n = 6$ rats per group, two-tailed Mann–Whitney U-test). **\*\***$P < 0.01$; **\*\*\***$P < 0.001$. For each box plot, center bar indicates median, box indicates 25th and 75th percentiles, and whiskers extend to the most extreme data points without outliers ($\pm 1.5$ IQR). Outliers are shown in dot signs. Bar graphs depict data as mean $\pm$ SD. WF-stim wide-field stimulation, SNF-stim sequential narrow-field stimulation. For detailed statistical information, see Supplementary Table 1. Source data are provided as a Source Data file.

These findings suggest that the SNF stimulation can effectively inhibit excessive neuronal activation and prevent neuronal loss during KA-induced SE. However, the seizure-suppressing effect had no particular correlation with the increase in GABAergic neuronal activation in the present study.

## SNF stimulation in a chronic model with spontaneous recurrent seizures

While the superiority of the proposed SNF modality including the anti-seizure effect and safety with regional specificity was demonstrated via electrophysiological and biological investigations, there

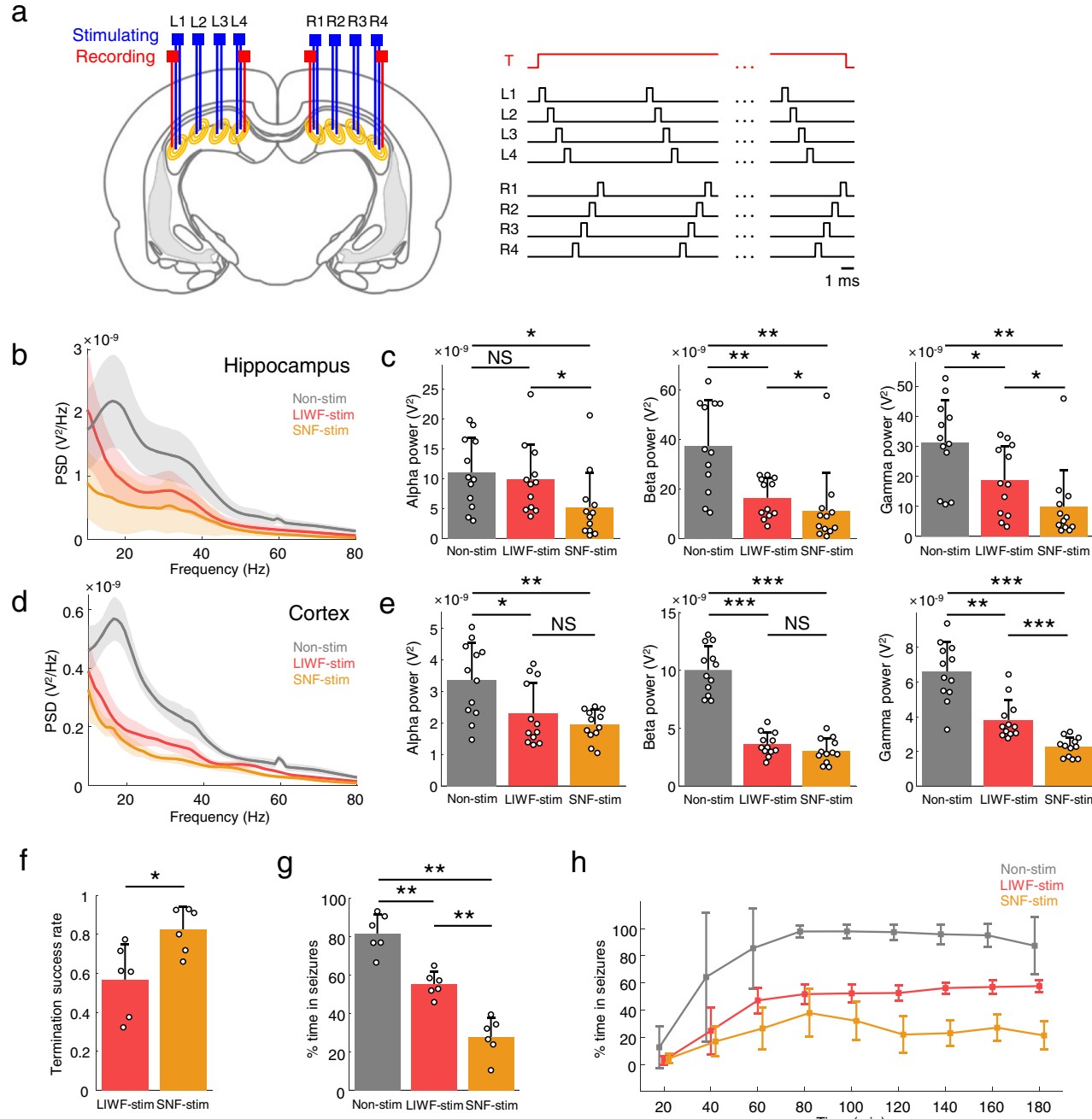

**Fig. 5 | Comprehensive analysis of SNF stimulation for suppressing seizures.**
**a** Schematic depicting the bipolar and tripolar depth electrodes implanted bilaterally for SNF stimulation (left). Experimental protocol for SNF stimulation (right). When a trigger signal is generated by the detection algorithm, current stimuli are sequentially applied to the electrodes. **b**–**e** Comparison of power spectra.
**b, d** Power spectral density (PSD; Welch's method with 1 s windows and 0.5 s overlap) for the non-stimulation (gray), LIWF stimulation (coral-red), and SNF stimulation (orange) groups. Lines represent the means and shaded areas represent SD. **c, e** Average power for the alpha, beta, and gamma frequency bands from the hippocampus (**c** P(alpha) = 0.019, P(beta) < 0.001, and P(gamma) = 0.0016, n = 12 in all comparisons, Kruskal–Wallis test with Bonferroni correction) and cortex (**e** P(alpha) = 0.01, P(beta) < 0.001, and P(gamma) < 0.001, n = 12 in all comparisons,

Kruskal–Wallis test with Bonferroni correction) channels, respectively. **f** Probability of terminating seizures with LIWF and SNF stimulation (P = 0.0202, n = 6 rats per group, two-tailed Mann–Whitney U-test). **g, h** Comparison of percentage times in seizure episodes for non-stimulation, LIWF stimulation, and SNF stimulation groups during total sessions (**g** P < 0.001, n = 6 rats in all comparisons, Kruskal–Wallis test with Bonferroni correction) and segmented sessions (**h** n = 6 rats in all comparisons). *P < 0.05; **P < 0.01; ***P < 0.001; NS, not significant. Bar graphs depict data as mean ± SD. Data for non-stimulation from the same subjects as shown in Fig. 3. Non-stim non-stimulation, LIWF-stim low-intensity wide-field stimulation, SNF-stim sequential narrow-field stimulation. For detailed statistical information, Supplementary Table 1. Source data are provided as a Source Data file.

still remains questions in establishing solid clinical and translational relevance for human TLE due to the use of an acute SE model and anesthetic agent. To answer these questions, experimental validation of the SNF stimulation was further investigated in an awake rat model with spontaneous recurrent seizures (SRSs). We first modeled

chronic TLE rats with KA injections and implanted custom-made portable brain stimulators into the epileptic rats to comprehensively investigate the both therapeutic and possible adverse effects during SNF stimulation (Fig. 8a, b, Methods). The seizure-terminating effects of SNF stimulation were comprehensively investigated by comparing

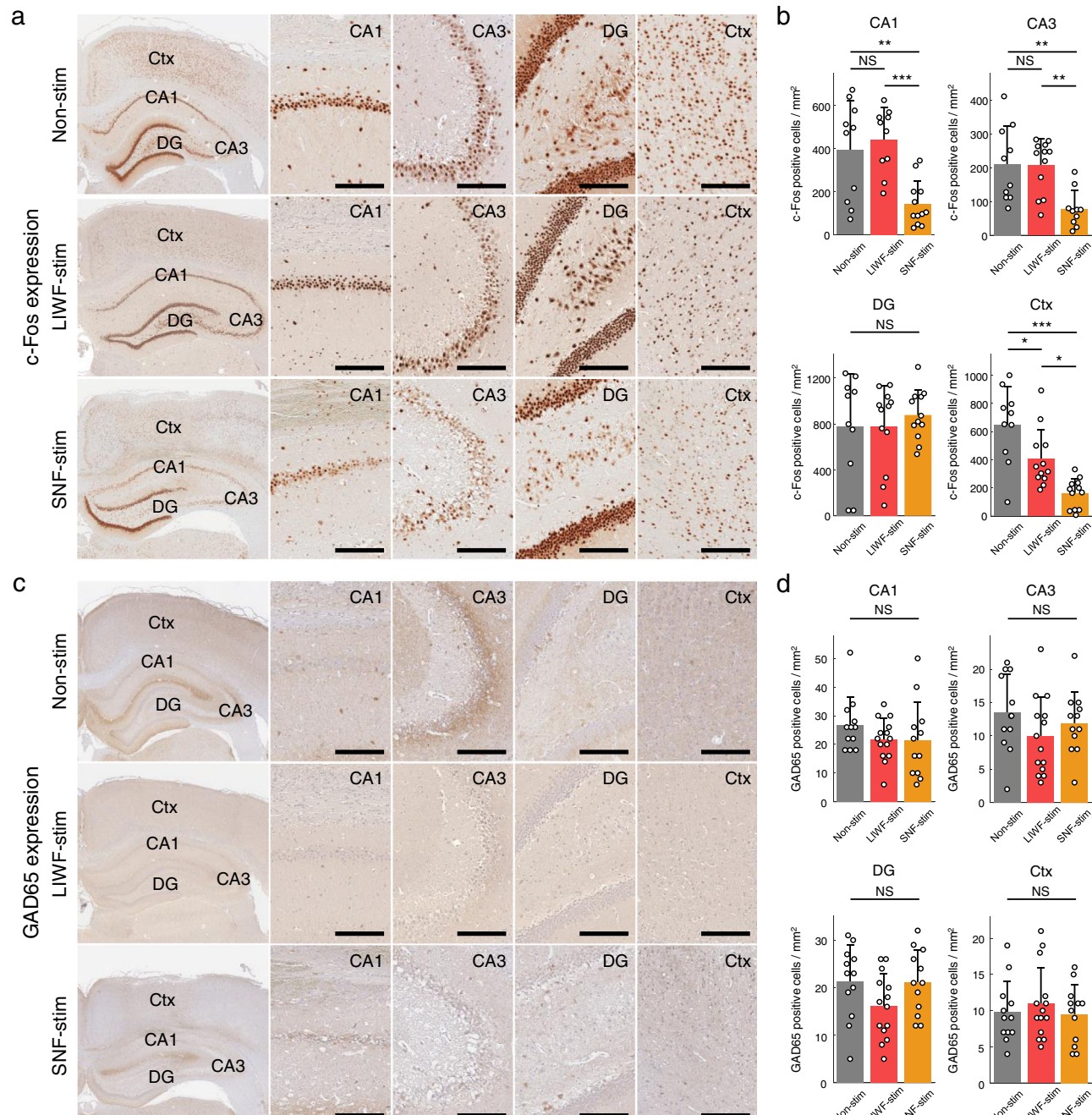

**Fig. 6 | Analysis of the therapeutic effects of SNF stimulation in biological levels. a–d** Comparisons of c-Fos (**a**, **b**) and GAD65 (**c**, **d**) labeling in the hippocampus and cortex with non-stimulation, LIWF stimulation, and SNF stimulation in an acute SE model. **a**, **c** Representative brain sections of the KA-induced acute seizure model labeled with c-Fos and GAD65 ($n = 6$ rats per group). Magnified images show gene expression in detail in the specific regions including the CA1, CA3, DG, and cortex, respectively. **b** Quantification of c-Fos positive cells in the unit area after non-stimulation, LIWF stimulation, and SNF stimulation treatment ($P = < 0.001$, 0.0014, 0.32 and $< 0.001$ for CA1, CA3, DG, and cortex each, $n = 10$–12 samples in all comparisons: precise values in Supplementary Table 1, one-way ANOVA with Bonferroni correction). **d** Quantification of GAD65 positive cells in the unit area after non-stimulation, LIWF stimulation, and SNF stimulation treatment ($P = 0.3426$, 0.2648, 0.1138, and 0.687 for CA1, CA3, DG, and cortex each, $n = 12$–14 samples in all comparisons: precise values in Supplementary Table 1, one-way ANOVA with Bonferroni correction). *$P < 0.05$; **$P < 0.01$; ***$P < 0.001$; NS, not significant. Bar graphs depict data as mean ± SD. Scale bars in **a** and **c**, 200 μm. Non-stim non-stimulation, LIWF-stim low-intensity wide-field stimulation, SNF-stim sequential narrow-field stimulation. For detailed statistical information, Supplementary Table 1. Source data are provided as a Source Data file.

the non-stimulated, LIWF-stimulated, and SNF-stimulated rats. The seizure duration was significantly reduced during SNF stimulation compared with non-stimulation and LIWF stimulation groups (Fig. 8c, d; 7.5 (IQR = 3.25–9.5), 19.75 (IQR = 12.44–26.75), and 28.88 (IQR = 22.25–33.88) s for SNF-stimulated, LIWF-stimulated, and non-stimulated comparison; $P < 0.001$ for one-way ANOVA; $P < 0.001$ for

Kolmogorov–Smirnov test in all comparisons). To test whether hippocampal SNF stimulation could affect normal rodent behavior or elicit anxiety-related behavior in a chronic model, we conducted the place preference test and open field test (Fig. 8e–i). In both experiments, no significant difference was observed between the non-stimulation and various stimulation intensity conditions during SNF

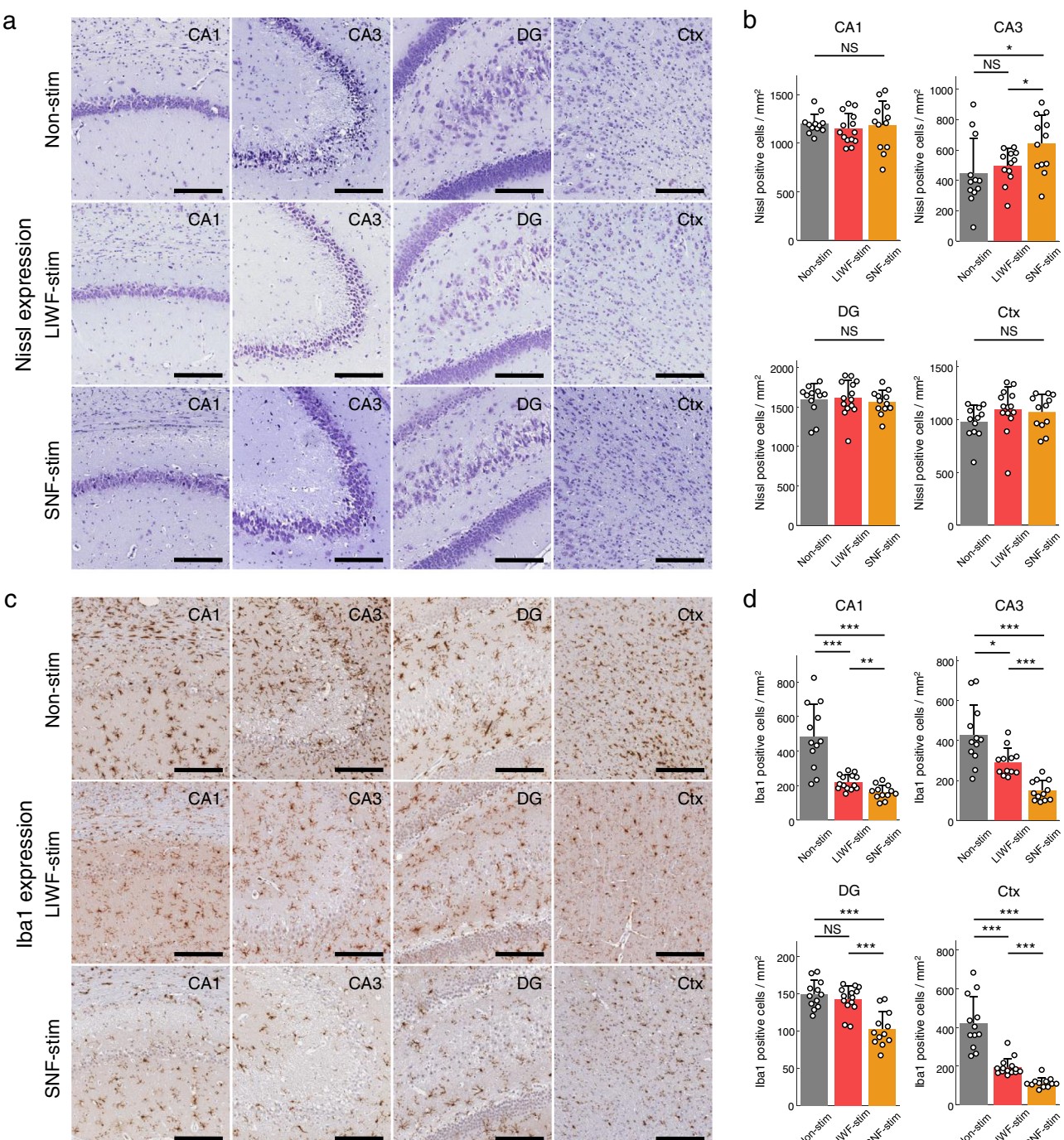

**Fig. 7 | Evaluation of neuronal damage and neuroinflammation. a–d** Analysis of Nissl staining (**a**, **b**) and Iba1 immunoreactivity (**c**, **d**) in the rodent brains after acute seizure induction. **a**, **c** Representative histological sections stained with Nissl and Iba1 of the brain regions including the CA1, CA3, DG, and cortex (*n* = 6 rats per group). **b** Quantification of Nissl positive cells (intact neurons) for the non-stimulation, LIWF stimulation, and SNF stimulation groups (*P* = 0.7833, 0.0392, 0.253, and 0.7673 for CA1, CA3, DG, and cortex each, *n* = 12–14 samples in all comparisons: precise values in Supplementary Table 1, one-way ANOVA with Bonferroni correction). **d** Quantification of microglial activation for the non-stimulation, LIWF stimulation, and SNF stimulation groups (*P* < 0.001 for all comparisons, *n* = 12–14 samples in all comparisons: precise values in Supplementary Table 1, one-way ANOVA with Bonferroni correction). **P* < 0.05; ***P* < 0.01; ****P* < 0.001; NS, not significant. Bar graphs depict data as mean ± SD. Scale bars in **a** and **c**, 200 μm. Non-stim non-stimulation, LIWF-stim low-intensity wide-field stimulation, SNF-stim sequential narrow-field stimulation. For detailed statistical information, Supplementary Table 1. Source data are provided as a Source Data file.

stimulation, whereas significant differences were observed in the WF stimulation group (Fig. 8f, h, i). Taken together, these experimental results demonstrate that SNF stimulation can effectively inhibit spontaneous seizures in the brain of a chronic TLE model while preventing undesired off-target neuronal modulation, suggesting the clinical and translational relevance of the SNF modality for human trials.

## Discussion

In this work, we present a new hippocampal stimulation method called SNF stimulation, which effectively terminates temporal lobe seizures using sequentially organized micro-stimulation while preventing fringing field effects that can evoke multiple side effects. The effectiveness of this new epilepsy control method is shown through a rigorous and systematic approach, revealing the therapeutic effect of

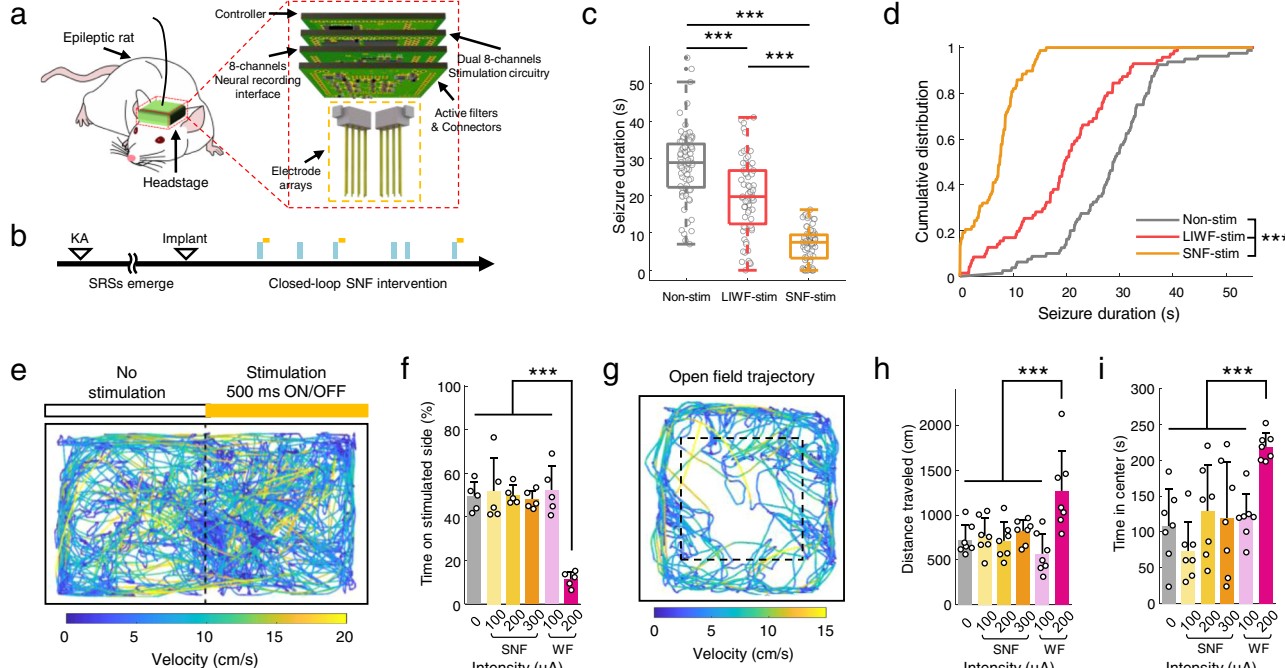

**Fig. 8 | Chronic seizure control and cognitive-behavioral responses during SNF stimulation. a** In vivo chronic experimental setup with implantable stimulator. **b** Experimental timeline. **c** Seizure duration in non-stimulation ($n = 80$ sessions), LIWF stimulation ($n = 71$ sessions), and SNF stimulation ($n = 78$ sessions) groups using a chronic seizure model ($P < 0.001$, one-way ANOVA with Bonferroni correction). Center bar indicates median, box indicates 25th and 75th percentiles, and whiskers extend to the most extreme data points without outliers (±1.5 IQR). Outliers are shown in dot signs. **d** Corresponding cumulative distributions of seizure duration ($P < 0.001$ for all comparisons, Kolmogorov–Smirnov test). **e** Example trajectory of a rat model with SRSs during place preference test in SNF condition. Orange bar indicates stimulation zone. Traces are color-coded for velocity.

**f** Percentage times spent on stimulation side at varying stimulus intensities ($P < 0.001$, $n = 5$ per group, one-way ANOVA with Bonferroni correction). **g** Example path-tracing during SNF stimulation in an open field. Travel distance (**h** $P < 0.001$, $n = 7$ per group, one-way ANOVA with Bonferroni correction) and time in center region (**i** $P < 0.001$, $n = 7$ per group, one-way ANOVA with Bonferroni correction) during stimulation. ***$P < 0.001$; NS, not significant. Bar graphs depict data as mean ± SD. Non-stim non-stimulation, WF-stim wide-field stimulation, LIWF-stim low-intensity wide-field stimulation, SNF-stim sequential narrow-field stimulation. For detailed statistical information, Supplementary Table 1. Source data are provided as a Source Data file.

overall hippocampal stimulation and the results of comparing the electrophysiological responses induced by various stimulation configurations. Using WF stimulation, which is a conventional electrical DBS modality with applying stimulus current broadly to target region, in a rat model of TLE, we found that stimulation applied broadly to the entire region of the bilateral hippocampal structure effectively inhibited ictal activity. However, we also found that the WF stimulation elicited unwanted neural responses in neighboring tissue even with a several-fold smaller stimulus current compared with SNF stimulation. In this context, we established the safety threshold as the stimulus intensity that does not evoke undesired off-target neuronal activation or inhibition to investigate the translational relevance of SNF stimulation for human TLE. In SNF stimulation at its safety level, the sequentially induced fields effectively disrupted the epileptic rhythms and enabled the SE animals to stop experiencing electrographic seizures, whereas the LIWF stimulation (WF stimulation with its safety limit) did not show reliable anti-seizure effects. These findings suggest that the proposed SNF stimulation can efficiently reduce the undesired stimulation of adjacent brain tissue, while effectively alleviating epileptic seizures in TLE as a spatially elaborate stimulation modality.

The rodent hippocampus exists in a bilateral structure and lies at the center of the brain symmetrically like humans, and its substantial neural activities can propagate bilaterally along the dorsal and ventral commissural pathways[47–49]. Numerous investigations aimed at ameliorating epileptic seizures with electrical and optical stimulation have been conducted using unilateral stimulation, both ipsilateral and contralateral to seizure-inducing sites[17,50,51]. Some studies have investigated neuronal synchrony in the epileptic brains of humans and

rodents, and revealed an increase in synchrony as a typical feature of the ictal state[34,52,53]. In this context, we hypothesized that the unilateral intervention may not effectively alleviate the hyper-synchronized rhythms in the hippocampus if the unilateral modulation does not reach the level required for modulating the contralateral rhythms. Our in vivo experiments were conducted to verify this assumption, and the results showed that the bilateral hippocampal stimulation significantly desynchronized the ictal network compared with the unilateral stimulation, resulting in ~2 times superior seizure-suppression effects.

In the rodent brain, the cerebral cortex is one of the regions adjacent to the hippocampus[54], thus we compared the fringing field effects that could lead to undesired neuronal responses evoked by two different stimulation methods via probing the induced field distribution in the cortex, estimating the neural responses in the M1 region with a simulation model, and measuring motor responses in freely moving rats. Taken together, these results demonstrate the superiority of the SNF stimulation method, which prevents unwanted neuronal activation outside the target site of a rodent TLE model. However, the nature of a more deeply located and elongated human hippocampus may imply more complicated and diverse adverse effects brought on by the evoked fringing fields during hippocampal stimulation. Considering the highly linked vicinity of the hippocampus proper – the medial amygdala[21], the entorhinal cortex[55], the posterior cingulate gyrus[56], and the remotely anterior thalamic nucleus[57] via the Papez circuit, DBS-related side effects, such as memory impairment, emotional derangement, and sleep disturbances may ensue as a result of orthodromic or antidromic wide-spreading fields in clinical scenarios[58,59]. For this reason, we propose the SNF stimulation method as an appropriate

approach for preventing possible side effects induced by spreading fields during overall hippocampal control, while concurrently maintaining effective epileptic control. With in vivo and in silico approaches, we demonstrated the superiority of the SNF modality that can concentrate stimulation fields on the target site (i.e. hippocampus), however, since the hippocampus and its neighboring structures are not only located close together, but also connected to each other via nerve axons, the stimuli delivered to the hippocampus might indirectly modulate adjacent tissues via axonal pathways. Thus, further studies determining responses mediated indirectly via axonal pathways will confirm the safety and feasibility of SNF stimulation for human TLE.

Hippocampus is widely accepted as responsible for declarative and emotional memory, thus we cannot avoid considering that direct hippocampal control may affect these functions and eventually lead to adverse effects. In previous studies, side effects except infection and hemorrhage were rarely reported[60], but a clinical trial of amygdala-hippocampal stimulation for TLE treatment reported reversible memory impairments in patients under high voltage stimulation[61]. However, the proposed SNF modality requires a relatively small intensity to induce field distributions sufficient to produce therapeutic effects compared to conventional techniques and the on-demand stimulation scheme delivers stimulus energy only at the event, thus it is considered that hippocampal SNF stimulation itself is unlikely to produce side effects. Nonetheless, these concerns should be further clarified through human clinical studies at the human physiology and psychology level for safer clinical translation of SNF methods.

As a potent analog of glutamate, KA has been used to induce intensive depolarization and subsequent cell death, which is a central phenomenon of TLE[62,63]. In the present study, histopathological examinations in brain sections of the acute SE model with antibodies for the neuronal and synaptic activity marker c-Fos[64] and GAD65[65], the neuronal marker Nissl[66], and the microglial marker Iba1[67] were conducted to investigate the epileptic network responses induced by the seizure-terminating effects of SNF stimulation over LIWF stimulation. The significant decrease in c-Fos expression in rats treated with SNF stimulation signaled a strong inhibitory effect to neural hyperactivation by excitotoxicity during SE, and this phenomenon ultimately resulted in substantial decreases in neuronal damage in the CA3 region, observed via Nissl expression. In Iba1 analysis, a significant decrease in the activation of microglia after SNF stimulation was found, implying that possible morphological and functional changes in response to brain injury in the experimental epilepsy model may have been prevented or suppressed with the proposed SNF stimulation[68]. Moreover, these histological results are consistent with the suppression effects confirmed by the LFP signal of electrographic seizures in the SNF stimulation group compared with non-stimulation and LIWF stimulation ones. Although the obvious electrophysiological and biological anti-seizure effects of SNF stimulation were shown via our findings, no difference was observed between all groups in the expression of GAD65, which can respond to short-term increases in the demand for GABA, a major inhibitory neurotransmitter[69], the expression of which is predominant in the GABAergic synapse for tonic inhibition regulating epileptiform activity[70]. With regard to these findings, we postulate that the seizure-alleviation effects of closed-loop SNF stimulation do not derive from augmenting the neural network in a GABAergic inhibitory fashion. In this context, two network mechanisms can be considered as the rationale for the observed seizure-inhibitory effects. The first network mechanism that could underlie the oscillation in epileptic networks could be disrupted by the different oscillation frequencies from SNF stimulation. Neuronal networks have intrinsic oscillation frequencies, and epileptic networks also have their own resonating characteristics[71]. If the SNF stimulation superimposes a different oscillatory pattern on pre-existing epileptic oscillations, the newly added one could disturb the original one[72], resulting in a reduction of epileptic oscillation. The second is the state

transitions by strong perturbations on the network dynamics in paroxysm[73]. Each narrow-field stimulation activates only a small region in the relatively large hippocampal structure and thus may not be sufficient for modulating the entire epileptic circuits, however, sequentially applied gradients could be precisely integrated into the target region, thereby superimposing their impacts spatiotemporally. In addition, it was confirmed that localized fields induced in random order could not terminate the epileptic rhythms properly (Supplementary Fig. 8), suggesting that it is important to sequentially organize the micro-stimulation to achieve anti-seizure effects successfully. As a result, the network dynamics in paroxysmal states could be collapsed by the timely intervention of SNF stimulation. Therefore, SNF stimulation deserves additional mechanistic investigation and safe translation at the cellular or network level for human use in the future.

Although the field distribution was restricted to the hippocampal region, the SNF stimulation significantly affected cortical rhythms and eventually suppressed diffuse seizures that had been fully propagated to the cortex as well as hippocampal seizures. This result implies that hippocampal SNF stimulation might deliver the stimulus energy via an extrahippocampal connecting pathway, resulting in strong seizure-suppressing effects even in diffuse seizures. Furthermore, the reduction of c-Fos activation in the cortex after stimulation agrees well with the suppression of electrographic seizures and attenuation of spectral densities during stimulation. However, SNF stimulation did not reduce c-Fos expression in DG even though electrographic seizures were suppressed. In this context, we can consider that closed-loop SNF stimulation may not be able to inhibit the biological response of potently acting KA in DG, even if it had consistently suppressed seizure activities as reported in previous studies[74–76]. Even if the above claims about the experimental results are reasonable, further studies at the cellular level are needed to confirm the exact rationale for these phenomena.

In the present study, acute and chronic seizure rodent models were used to reproduce the key features of human TLE. However, there is no experimental animal model that can fully reproduce all characteristics in human TLE patients including neuropathological, electroencephalographic, and behavioral features[27,77]. In particular, the acute SE model typically has seizure activity characterized by short duration and excessive recurrence, which are somewhat different symptomatic features compared to those of human TLE[78]. Thus, further studies with various experimental models[78–80] in acute and chronic trials will raise the assurance of the validity of translational relevance of SNF stimulation for human TLE patients.

Recently, electrode arrays for directional DBS have been introduced as an advanced technology that has the potential to improve the capabilities of DBS while preventing stimulation-induced side effects[81–83]. The current steering method for directional stimulation can be achieved by controlling the channel activation of multi-channel electrodes and thus is capable of rendering micro-stimulation. However, in existing directional steering technologies, as the target area becomes longer, the induced field tends to spread more broadly[82,84], thus stimulation targets with a relatively large and longer structure, such as the hippocampus, present limitations for precisely controlling the volume of tissue activated (VTA). While the emerging current steering technique is still limited in terms of its application to overall hippocampal control due to challenges mentioned above, our proposed SNF stimulation indicated precise control for an elongated tissue structure and can thus be considered as a future modality for closed-loop hippocampal stimulation methods. More specifically, in SNF stimulation, multiple localized fields are induced to synchronously modulate the neuronal network thus each activation channel can be overlapped freely in sequence to drive the desired field distribution, resulting in the fields can be formed much further precisely in terms of spatial resolution, compared to conventional ones. Another advantage of the proposed method is that it can be implemented using

existing directional electrodes or conventional multi-channel electrodes, such as SureStim1 (Medtronic, USA), 6180 (St. Jude Medical, USA), and µDBS[85,86] by implementing switching circuitry in the DBS pulse generator. Accordingly, SNF stimulation can achieve more tailored neuromodulatory effects. Further studies on stimulation-induced field intensity, shape, and VTA in a chronic animal model of epilepsy and human case studies will confirm the clinical feasibility of SNF stimulation.

Developing a fully automated closed-loop stimulation method with the proposed system for the treatment of epileptic seizures has gained a particular interest due to its following advantages. First, stimulation in a timely manner before seizures being fully developed can easily abort the epileptic rhythms[87,88]. Second, on-demand stimulation delivers energy only at the onset of the event, thereby avoiding adverse effects elicited by the disruption of normal brain activities compared with continuous or cyclic open-loop stimulation[89,90]. Third, on-demand intervention can prevent stimulation tolerance, which current open-loop paradigms may contribute to[91,92]. Furthermore, recent advancements in artificial intelligence (AI) techniques, such as deep learning and reinforcement learning, can be applied to parametric optimization including stimulus intensity, width, repetition rate, and duration of an on-demand stimulation strategy to enhance its automaticity and subsequent therapeutic adequacy[93,94].

In conclusion, we propose the SNF stimulation method for the treatment of spontaneous recurrent seizures in TLE while performing the precise control of neuronal circuits, and demonstrate its therapeutic efficacy through comprehensive in vivo and in silico studies. Our findings suggest that the proposed SNF stimulation can be further expanded to control other drug-refractory neurological disorders through its capability of precisely and selectively modulating the target brain region.

## Methods

### Animals
All experimental investigations were approved by the Institutional Animal Care and Use Committee (IACUC) at Pohang University of Science and Technology (approval number: POSTECH-2020-0083) and Samsung Biomedical Research Institute (approval number: 20220509001), and were conducted in compliance with the National Institutes of Health Guide for the Care and Use of Laboratory Animals. Sprague Dawley male rats (250–350 g, 8 weeks old, ORIENT, Korea) were used in this study.

### Design and implementation of the closed-loop feedback stimulation system
The developed closed-loop feedback stimulation system comprised a recording module, a stimulation module, and custom LabVIEW (National Instruments, USA) program for computation (Figs. 1a and 8a). A biosignal sensing analog front-end chip (ADS1299, Texas Instruments, USA) and active filter circuitry (implemented with AD8508, Analog Devices, USA) were adopted as a measuring device for LFP recording. This module provides simultaneous recording for 8-channels (up to a 16 kS/s sampling rate for each channel with 24-bit resolution and a programmable gain amplifier). The isolated current source in the stimulation device primarily comprised an analog-to-digital converter (DAC8580, Texas Instruments; AD5420, Analog Devices), an operational amplifier (MC34074, ON Semiconductor, USA), and several analog multiplexer chips. This stimulator was able to drive voltages up to 14 V with a maximum amplitude of 3 mA and an arbitrary waveform. Both modules were connected to a PC using a microcontroller unit (SAM3X8E, Microchip Technology, USA), which provides DMA to enable high-speed USB communication. The recorded LFP was processed using custom LabVIEW program to detect electrographic seizures in real-time. When a seizure was detected, a trigger signal was delivered to the stimulation module.

### Surgical procedure for implantation of the stimulating and recording electrodes
Rats were anesthetized with 4% isoflurane initially and with 1–3% isoflurane in the prone position under a stereotaxic apparatus (68002, RWD Life Science, China) during surgery. The body temperature of the anesthetized rats was maintained at 37 °C during surgery with an electronic heating pad. A small midline incision was made to expose the skull, then the skull's surface was cleaned and dried. The bipolar and tripolar electrodes were custom-made with 200 µm diameter polyester-insulated stainless steel wires (Goodfellow, UK), and their end-tips were exposed with 400 µm spacing vertically. Depending on the type of stimulation, different numbers of burr holes were drilled (Figs. 2a, 3a, and 5a). In all animals, six burr holes were made; two for the reference and ground screw electrodes in the left and right frontal bones, respectively; two for the anchor screws in the left and right parietal bones near the coronal bone suture; two for cortex recordings in the front of the left and right lambdoid sutures (all screws had a 1.2 mm diameter). For the non-stimulation and stimulation groups, except for the SNF stimulation, four custom-made bipolar depth electrodes were bilaterally implanted in the hippocampal region for stimulating and recording (−2.2 mm AP, ±2.2 mm ML, −3.6 mm DV, and −4.5 mm AP, ±4.5 mm ML, −4.0 mm DV). For the SNF stimulation group, four bipolar and four tripolar electrodes were implanted into the hippocampus using the custom-made 3D-printed frame for supporting the depth electrodes (bipolar: −3.0 mm AP, ±2.3 mm ML, −3.6 mm DV, and −3.7 mm AP, ±3.4 mm ML, −3.6 mm DV; tripolar: −2.2 mm AP, ±1.2 mm ML, −3.6 mm DV, and −4.5 mm AP, ±4.5 mm ML, −4.0 mm DV). For the SNF stimulation, the bipolar electrodes were used for stimulation (one serving as an anode and the other as a cathode), and the tripolar electrodes comprised one pair for stimulation and one for LFP recording. Both the depth and screw electrodes were connected to the connectors and then fixed to the surface of the skull with dental cement. The implanted rats were housed individually during recovery periods.

### Preparation of SE model and chronic TLE model with KA injection
To reproduce the key features of human TLE, we used the KA-induced seizure model in both acute and chronic conditions. To implement an acute SE rodent model, rats were first implanted with the electrodes. Following recovery from the implant surgery, the rats were initially anesthetized with 4% isoflurane in the induction chamber. After inducing anesthesia, ketamine (80 mg/kg) was intraperitoneally injected and isoflurane was discontinued to remove the effect of isoflurane anesthesia on neuronal activity. Then, the rats received the KA (Hello Bio, UK) by intraperitoneal injection at a dose of 7 mg/kg of body weight, dissolved in 5 mg/mL of normal saline. Finally, the rats were moved to the plastic recording cages and connected to the recording and stimulating system for acute experiments.

For chronic seizure induction, non-implanted rats were moved to a single cage and then intraperitoneally injected with KA (5 mg/kg) every hour, until convulsive motor seizures during SE were observed[95–98]. A few hours after SE, all rats were subcutaneously administered with normal saline and given moistened rat chow, then offered a regular diet with monitoring for 4–8 weeks. Rats that developed SRSs were selected for chronic condition experiments and then electrode implantation was performed. After 2-weeks recovery period, the rats were moved to the experimental cages and connected to the closed-loop stimulation system.

### Recording and processing the electrophysiological activity for electrographic seizure detection
The LFP from both hippocampi (4-channels) and cortex (2-channels) were recorded with a sampling rate of 500 Hz and 1× gain. The signal was passed through a 0.5 Hz high-pass filter to remove the baseline wander. The filtered signal was processed one frame at a time (each

frame consisted of 250 samples (0.5 s with 50% overlap). In the acute experiments, the baseline for each subject was computed for 5 min after seizure induction. Seizures were identified using criteria where a power spectral density in a specific band (alpha, beta, and gamma band, respectively) was greater than twice the baseline[99,100]. A stimulus trigger was generated when any of the channel thresholds were met. For the chronic model, both frequency properties and spike features were used to detect electrographic seizures while preventing false positives due to motion artifacts in freely moving rats. The power changes in specific frequency bands, which was used to detect seizures in the acute trial, were computed with the function that actively varies the trigger criteria based on baseline signal in real-time. Spike characteristics including amplitude, rate, and regularity were calculated with fine-tuning for each individual and used to determine trigger outputs[50,101,102]. In addition, abnormal patterns such as abnormally large amplitude signals and certain repetitive features due to external artifacts were excluded from the detection process to prevent false alarms. These procedures were equally performed for post-analysis using custom MATLAB (Math-Works, USA) scripts to compute factors including the total seizure duration, and the success rate of termination. The phase synchrony index was calculated as the phase difference between both hippocampi recorded (Fig. 3b–e and Supplementary Fig. 2a; index $R = |<e^{i\Delta\theta}>|$, $\Delta\theta$ is the phase difference between two signals)[103–105].

### Induced electric field design for SNF stimulation

Prior to designing and optimizing the electrode configuration, the effect of irregularities in the electrical properties of tissues on the electric field formation was first investigated using Sim4Life (Zurich MedTech AG, Switzerland), which is specialized in simulating electromagnetic (EM) propagation in a biological structure. The induced field distribution when tissue has heterogeneous electrical conductivities was compared with that of homogeneous condition (Supplementary Fig. 9a, b). The conductive values in heterogeneous conditions were randomly determined with an error within 10% of the homogeneous condition (0.7 S/m). The results showed that there was no significant difference in the induced gradients, lower than 10%, implying that the heterogeneous nature of the tissue in real-world cases would not significantly affect the designed output of field formation under homogeneous condition (Supplementary Fig. 9c, d).

Next, we pre-defined that the stimulation pulse needs to induce an electric field of at least 100 mV/mm in the target tissue, the hippocampus, to suppress epileptic rhythms[36,44] while minimizing off-target spreading fields. No consideration of field orientation for each electrode was given, based on the results of previous studies[36]. In our pilot test, we confirmed that delivering stimulation to the ventral hippocampus did not show significant seizure-inhibitory effects like those of the dorsal hippocampus, and the direction of electrode activation, from dorsal to ventral or vice versa, did not affect the therapeutic effects, thus the experimental protocol was designed to modulate the dorsal hippocampal region with the stimuli from ventral to dorsal direction (note that the direction of applying the spatiotemporally organized electric field was not important, but the unorchestrated field that was induced in random order did not function properly; see Supplementary Fig. 8). EM simulations with varying the position and number of electrodes were performed in compliance with the above considerations (Supplementary Fig. 1). The results showed that as the number of electrodes increased, the stimulus intensity required to inhibit seizures decreased, while the absolute value of the decreasing slope gradually decreased (Supplementary Fig. 1b). Also, it was confirmed that the off-target spreading fields rapidly decreased with the increase in the number of electrodes in SNF stimulation, but this effect was finally plateaued when there were >4-pairs (Supplementary Fig. 1c) and additional electrodes were not justified anymore. Thus, a 4-pairs configuration was adopted, which relatively less causes tissue damage due to electrode insertion and can induce a field sufficient to suppress

epileptic networks while minimizing off-target fringing fields, with relatively low stimulus intensity.

### Stimulation parameters

Based on the report that high-frequency stimulation provides anticonvulsant effects, 130 Hz charge-balanced biphasic current pulse stimulation was adopted in this study[10,106]. The pulse width of the stimulus was 300 μs for each (total biphasic pulse width of 600 μs), and no interphase gap. The stimulus pulse was applied for 5 s for each stimulation. The stimulus amplitude was determined as the minimum value indicating anti-seizure effects for each subject via intensity titration with steps of 50 μA (up to 650 and 140 μA for SNF and LIWF stimulation, respectively).

### Induced electric field measurements

To record the stimulation-induced electric field distribution, rats were anesthetized with isoflurane and placed in a stereotaxic frame as described in the above surgical procedure. After the skin was retracted with a midsagittal incision, 54 holes were drilled on the skull throughout the whole brain region for measurement and four holes were drilled in the right hemisphere for stimulation. Custom-made recording electrode arrays (1.5 mm spacing) were inserted at 54 distinct positions in the brain, 1 mm deep below the dura. Four bipolar electrodes for stimulation were implanted using a 3D-printed frame in a similar manner as the montage for SNF stimulation. Each electrode was made by a polyester-insulated stainless steel wire (200 μm diameter). A needle electrode was inserted into the thigh and served as a reference electrode for recording intracerebral electric potentials. Then, the recording electrodes were connected to a NI-9220 data acquisition (DAQ) device (National Instruments), and stimulating ones were connected to a custom-made stimulator as described before[107].

10 Hz biphasic pulses were delivered through paired electrodes in five configurations (Fig. 4a; WF-stim; R1, R2, R3, and R4 for SNF-stim). The induced electric potentials were recorded for 5 s with no analog filter at a 100 kHz sampling rate per channel. The recorded potentials were averaged for each condition, and then peak amplitude for each channel was used to calculate the induced electric fields. Similar to the protocol for measurements in entire brain regions, the linear characteristics of spreading fields were further investigated in the left hemisphere using a 10-channel recording electrode array (0.5 mm spacing).

### Modeling of the stimulation-induced neural responses

A simplified 3D model of a rodent brain was constructed with the 'Waxholm Space Atlas of the Sprague Dawley Rat Brain' (1024 × 512 × 512 voxels with 39 μm resolution)[108]. Tissue segmentation was conducted by iSEG platform (Zurich MedTech AG) with several image processing such as interpolation and outline correction. Tissue segmentation information including cortex, and hippocampus was imported into Sim4Life. Conductivity values were set to 0.0626 (for the brain, excluding the hippocampal region), 0.0988 (the hippocampus), $6.62 \times 10^{-7}$ (polyester), $1.45 \times 10^{6}$ (stainless steel), and 0 (air) S/m, respectively[109]. Next, simulations were performed to compute the electric fields induced by a stimulus of 1 mA in both WF and SNF stimulation configurations using the 'Quasi-static LF solver'.

The cortex somatosensory neuron models (L1 and L2) from the Blue Brain Project (BBP)[40,41] were implemented into Sim4Life. The model data in the original BBP, comprised of "hoc" files, were customized to allow for their importation into the simulator. The morphologies and coordinates of the imported neuron models were visualized with the automatic 3D reconstruction internal tool in Sim4Life based on NEURON[110] and the positions of neuron models were arranged with GUI tools of Sim4Life, placing them in the motor cortex of the customized 3D rat model.

A total of 18 neurons were placed on the M1 region in the 3D rat model. The numerical simulations were performed using the 'Neuron solver' in Sim4Life with the WF and SNF stimulation methods. The field intensity for inducing the action potential for each neuron and electrode montage was calculated using the 'titration procedure', one of the functions in the Neuron solver. All intensities were normalized to the computed threshold of each SNF stimulation.

## Comparing the threshold of the motor response in vivo

To compare the threshold of the motor response of WF and SNF stimulation, rats were prepared with configurations for WF and SNF stimulation. The rats were then moved to the experimental cage and connected to the stimulation system via a slip-ring connector to prevent twisting and excessive tension on the cables. We measured the stimulus amplitude that evoked an abnormal motor response (stages 4–5 on a modified Racine scale[111]) in freely moving rats by applying a stimulus for 1 s. The applied current was sequentially varied from 50 to 800 μA with a 50 μA step, and sufficient resting time was provided between each stimulus to prevent the cumulative effects of continuous stimulation.

## Experimental details for the acute in vivo stimulation test

All rats with KA injection in acute experiments were randomly divided into seven groups (*n* = 6 animals for each group); (1) non-stimulation group, (2) repetitive timing stimulation group, (3) closed-loop unilateral WF stimulation group, (4) closed-loop bilateral WF stimulation group, (5) closed-loop bilateral LIWF stimulation group, (6) closed-loop unilateral SNF stimulation group, and (7) closed-loop bilateral SNF stimulation group. All animals were monitored to detect electrographic seizures for 3 h after seizure induction. In the non-stimulation group, no action was taken except for monitoring and maintaining anesthesia. In the repetitive timing stimulation group, a stimulus current was delivered to both hippocampi every 5 min starting from 10 min after the seizure induction (Fig. 2a, b). In the closed-loop WF, LIWF, and SNF stimulation groups, the stimulation module delivered electrical pulses to the hippocampal region when it received a trigger signal from the custom seizure detector. Stimulation pulses were delivered only to the right hippocampus in the unilateral stimulation, whereas stimulation pulses were applied to both hippocampi in the bilateral configuration (Figs. 3a and 5a). In the SNF stimulation groups, pulses were sequentially applied to the right hippocampus or both hippocampi (Fig. 5a; L1, L2, L3, L4, R4, R3, R2, and R1, in that order; a 100 μs interval between each pulse). Stimulation was continuously applied until the seizure was terminated, and a non-stimulation interval of 10 s was set for each stimulus to prevent excessive stimulation. Electrical artifacts could be recorded and disrupt the computation for seizure detection during stimulation, thus the seizure detector was turned off when stimulation was in progress.

## Immunohistochemical analysis

At 3 h after the KA injection for acute experiments, rats were transcardially perfused with saline, and their brains were then dissected and fixed with 4% paraformaldehyde for 24 h at 4 °C. Brain tissue was embedded in paraffin and sectioned into 6-μm-thick coronal planes at −3.0 mm AP. To conduct histological analysis of each group, Nissl staining (Nissl Stain Kit, IHC WORLD, UK) and immunohistochemistry with antibodies including c-Fos (ab209794, dilution 1:200, Abcam, UK), GAD65 (ab26113, dilution 1:100, Abcam), and Iba1 (ab178846, dilution 1:2000, Abcam) were performed on brain sections. Prior to c-Fos staining, pH 6.0 citrate retrieval (ZUC028-500, Zytomed, Germany) treatment was performed at 121 °C for 7 min. Anti-mouse (K4001, dilution 1:2000, Agilent DAKO, USA) was used as the secondary antibody for c-Fos and GAD65, and anti-rabbit (K4003, dilution 1:2000, Agilent DAKO) was used as the secondary antibody for Iba1. The stained slides were scanned using an Aperio ScanScope AT system (Leica Biosystems, USA), and then captured the CA1, CA3, DG, and cortex area using the Aperio ImageScope software (Leica Biosystems).

Brain tissue with broken brain anatomical landmarks was excluded from the quantitative analysis. ImageJ (National Institutes of Health, USA) was used to measure the number of positive cells.

## Experimental details for the chronic experiments

Following the chronic seizure induction and electrode implantation procedures, animals were monitored 24 h a day to detect electrographic seizures with the algorithm that uses combinations of frequency and spike features. When the specified threshold for intervention was met, SNF or LIWF stimulation was immediately applied to the hippocampal region in each stimulated group (the non-stimulated group received no treatment). Stimulation was applied only once for each individual event, and a non-stimulation interval was given as 100 s.

## Place preference test

A 26 × 42 × 18 cm³ cage was divided into left and right areas of equal size. The rat was offered a period for psychological stability in the cage for 10 min, then treated with either SNF or WF stimulation when entering the stimulation zone, one of the two sides (repeated interval: 500 ms ON and 500 ms OFF). Stimulus current intensity was randomly assigned from among 0, 100, 200, and 300 μA in one session. Movement was tracked with a video camera positioned above the center of arena and the place preference ratio was calculated by dividing the time on the simulated side by the total time of the experimental session.

## Open field test

To investigate the induction of anxiety-related behavior during SNF and WF stimulation, rats were placed in a 40 × 40 × 20 cm³ arena and allowed to freely move around for 5 min. Stimulation intensity was randomly assigned in the same way used in the place preference test. Movement was tracked with a video camera located above the arena.

## Statistics and reproducibility

Data are presented as the mean ± standard deviation (SD) unless otherwise noted. Group comparisons were conducted using one-way analysis of variance (ANOVA), two-way repeated measures ANOVA or the Kruskal–Wallis test with Bonferroni correction for multiple comparisons. Two-tailed Mann–Whitney *U* test was used to compare two unpaired groups. Two-tailed student's paired *t*-test was used to compare pairwise data. Stimulation-induced electric field distributions for each configuration and chronic seizure durations were compared using a Kolmogorov–Smirnov test. For simplicity, *P*-values < 0.001 were reported as *P* < 0.001; otherwise, *P*-values were reported as an absolute value. A *P*-value below 0.05 was considered statistically significant. Statistical analysis was conducted using MATLAB with Statistics and Machine Learning Toolbox.

Experiments shown in Figs. 6a, c, and 7a, c and Supplementary Fig. 5c were repeated independently in at least four animals with similar results (exact numbers as indicated in figure legends, respectively).

## Reporting summary

Further information on research design is available in the Nature Portfolio Reporting Summary linked to this article.

## Data availability

The data supporting the findings in this study are available within the paper and its Supplementary information/Source Data file. The raw data are available for research purposes from the corresponding authors on request. Source data are provided with this paper.

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

## Acknowledgements

This research was supported by the following sources: the National R&D Program through the National Research Foundation of Korea (NRF) funded by the Ministry of Science and ICT (2021M3H4A1A03049084) (S.-M.P.), the Basic Science Research Program through the NRF funded by the Ministry of Science and ICT (NRF-2017R1A5A1015596) (S.-M.P.) and the Ministry of Education (2020R1A6A1A03047902) (S.-M.P.), the Pioneer Research Center Program through the NRF funded by the Ministry of Science, ICT & Future Planning (2022M3C1A3081294) (S.-M.P.), and a grant of the Korea Health Technology R&D Project through the Korea Health Industry Development Institute (KHIDI) funded by the Ministry of Health Welfare (HR21C0885) (Y.-M.S.).

## Author contributions

W.K., Y.-M.S., and S.-M.P. conceived and designed the study. W.K. designed and implemented the experimental system. W.K. and C.J. performed the in vivo experiments assisted by the clinical expertise provided by Y.-M.S and J.J. provided the technical and surgical support. W.K. and J.L. performed the numerical simulations. W.K. analyzed the data and prepared all figures. W.K., J.L., Y.-M.S, and S.-M.P. wrote the manuscript. Y.-M.S and S.-M.P. supervised the whole research.

## Competing interests

The authors declare no competing interests.
