## [Peer Review File · Nature Communications]

Closed-loop direct control of seizure focus in a rodent model of temporal lobe epilepsy via localized electric fields applied sequentiallyReviewers' comments:

Reviewer #1 (Remarks to the Author):

In this manuscript Kang and colleagues apply a closed loop neurostimulation strategy to disrupt acute seizures triggered by kainic acid. Conceptually this manuscript is built on a now long, and growing, literature on the use of closed loop neuromodulation strategies.

While the data presented appear sound, I have several problems with the over all design and some results.

First, the authors are examining acute kainate evoked seizures, as compared to spontaneous seizures occurring days/weeks after status epilepticus. The later is considerably more relevant to the human temporal lobe epilepsy condition.

Second, the animals in this study were anesthetized for the duration of the recording. This is confusing, and confounding. They are actually studying the stimulation on the background of ketamine anesthesia, which reduces the clinical and translational relevance.

Third, the representative seizures they show in Fig 2 suggest that their detection algorithm is sluggish. They are stimulating near the very tail end of seizures to begin with, making their effects less robust.

Fourth, the results in Fig 3D are not very robust. They analyze each bin with an independent Mann Whitney U test. A repeated measures ANOVA would be more appropriate. The independent values are not shown, making it impossible to really understand what the distribution is.

Fifth, there is basically no benefit to the sequential narrow field stimulation (SNF) over the wide-field stimulation in terms of stimulation efficacy, and the SNF stimulation was the major point of novelty for the manuscript.

Overall the manuscript presents some interesting data, but is limited in its novelty and represents a somewhat incremental advance to the field of neurostimulation for epilepsy. I would expect to see this in a more specialized journal (e.g., Brain Stimulation), as I do not think this is of interest to a broad audience.

Reviewer #2 (Remarks to the Author):

This manuscript reported a sequential narrow-field (SNF) stimulation method by applying sequential electrical stimulation on pairs of electrodes implanted in the hippocampus to control the localization of electric field. The authors compared the effects of open loop DBS, unilateral and bilateral closed loop wild field stimulation and SNF stimulation on the inhibition of seizure activities on an acute KA seizure model. The results showed the SNF stimulation method could achieve similar control effect similar as bilateral wild field DBS, but with less volume of infected brain area. This research is generally well designed and the manuscript is well organized. However, the idea of using multi implanted electrode to delivery small electric current to control the activation volume is lack of novelty, even this method is designed for seizure control. There are also several major questions need to be resolved:

1. the authors mentioned the SNF stimulation method could modulate the entire hippocampal region. However, the activate volume of electric current is very complex, not only depend on the parameters of electrical stimulation and the electrode shape, but also depend on the physiological characteristic of the targeted brain regions. There is not enough evidence to prove the modulation of entire hippocampus with SNF method by four pairs of parallel placed electrodes.

2. There might be other related brain nucleus activated by electrical stimulation in hippocampus, since the hippocampus has many adjacent structures and the current could preferentially activate the

passing axons from those adjacent structures. What's the effect of SNF stimulation of hippocampus on these adjacent structures in not well illustrated and explained.

3. In the experiment of comparison of firing field effects of SNF stimulation (Fig 5). The author measured the induced intracerebral electric fields on the contralateral motor cortex. Why not check the ipsilateral cortex and other related brain regions, such as those important area in papez circuit?

4. An acute status epilepticus (SE) model was used in this study. This model could activate frequent abnormal brain activities, but may not really reflect epilepsy situation of brain. The activation of SE state is not stable with animals and may fade away with short duration. It is better to test on a chronic seizure animal model. By the way, it is hard to do cognitive behavioral test on acute model. Although the authors mentioned less influence of SNF stimulation on normal cognitive behavior of animals, there is no evidence in the manuscript to prove this hypothesis.

5. In fig7a, compared with non-stim group, why the SNF stimulation caused more Nissl positive neurons in CA3 area but have no difference on the other stimulated areas? What's the effect of bilateral stimulation group?

Reviewer #3 (Remarks to the Author):

In this manuscript, Kang et al. investigate closed-loop control of temporal lobe seizures using a sequential narrow-field (SNF) stimulation method. This method is proposed to decrease the potential side effects of stimulation by reducing the region of brain tissue under the influence of the stimulation without compromising seizure termination efficacy. The motivation and concept for the work are interesting, but key points remain unaddressed and I have several questions regarding the analysis. Overall, further work is needed to improve understanding of this stimulation protocol, its efficacy, and potential for therapeutic applications. At present, the results presented are not sufficient to support the conclusions of the authors regarding the utility of this approach.

Major:

1) The innovation of the SNF method is purportedly its restricted gradient across adjacent brain structures. The data presented in Figure 5 incompletely characterize the method. Cortical sites in the contralateral hemisphere are presented for assessment of potential. Why were these sites selected? What about the ipsilateral hemisphere, where presumably the gradients are much higher? What about deep structures (such as thalamus or basal ganglia) that are in relatively close proximity to the hippocampus? How did the potential change over time with the sequential stimulation? The electric field is less for the SNF method, but is it still sufficient to drive/entrain action potential firing? Fields of 1mV/mm have been found to affect neural activity, so over what brain area could the SNF stimulation potentially change activity patterns? Motor responses were used to assess effects on these networks, but are these responses mediated primarily by cortex, or deep structures on the motor output pathway (e.g. basal ganglia or white matter pathways)? Overall, more detailed work to demonstrate the fields generated by the SNF and their effect on neural activity outside of the hippocampus are required.

2) The mechanism of the SNF in seizure termination is not directly investigated. This is a major gap in the present work. How many sites are required for this method? The authors use 4, but what determines this amount? What is the appropriate spacing – minimum and maximum? What about the orientation? Is it important that the sites activate sequentially in a particular order? The authors seem to orient the electrodes along the dorsoventral axis, perhaps in CA1 at each location, but this is not specified. Do the electrodes need to be activated from dorsal to ventral or vice versa, or is it directionally independent? The authors do not seem to survey ventral hippocampus. Is the seizure terminated throughout the hippocampus by the regional stimulation that they apply? Without this fundamental understanding of the method, it is difficult to appreciate its applicability and translational potential.

3) It seems as though there is a substantial lag between seizure onset and delivery of stimulation (see

Figure 1). In the example provided, the stimulation (orange) occurs at least > 15 s after the onset of the seizure. Was this duration standardized for all the seizures? Time prior to stimulation initiation could be a major confounder to overall seizure duration, as once the seizure has been established for longer it could be harder to stop.

4) The rationale for the histologic experiments is not clear to me. The SNF tissue is compared with unstimulated tissue, but the more apt comparison would be between the wide field (WF) and SNF stimulated tissue to determine whether the SNF tissue is less affected.

5) The methods for calculating the degree of synchronization, and what exactly this means, are not clear. It would be more accurate to compare individual seizure epochs to see whether seizures were expressed bilaterally and quantify this. The authors could separately look at synchronization of other hippocampal rhythms, such as theta, if of interest.

Responses to the reviewers' comments

We appreciate the reviewers for their careful review and insightful comments regarding our manuscript entitled "Closed-loop direct control of seizure focus in temporal lobe epilepsy via highly localized electric fields applied in a sequential manner." All the comments were very helpful in improving our manuscript, and we have addressed them in detail in the enclosed revised manuscript. Detailed point-by-point responses to the comments and corresponding changes are presented below.

Reviewers' comments

Reviewer 1:

In this manuscript Kang and colleagues apply a closed loop neurostimulation strategy to disrupt acute seizures triggered by kainic acid. Conceptually this manuscript is built on a now long, and growing, literature on the use of closed loop neuromodulation strategies. While the data presented appear sound, I have several problems with the overall design and some results.

Response: We appreciate the reviewer for the constructive feedback. We carefully reviewed the comment and provided the detailed responses as below:

1. First, the authors are examining acute kainate evoked seizures, as compared to spontaneous seizures occurring days/weeks after status epilepticus. The later is considerably more relevant to the human temporal lobe epilepsy condition.

Response: We appreciate the reviewer for this important comment regarding the translational relevance of the seizure model used in our study. We fully agree with the reviewer's comment that the chronic model is significantly more relevant to the real-world conditions of temporal lobe epilepsy compared to the acute model. As recommended, we conducted additional experiments using a chronic model with spontaneous recurrent seizures (SRSs). These new experimental results using SRS models also showed a significant seizure-suppressing effect of SNF stimulation and have been added to the Results section of the revised manuscript, "SNF stimulation in a chronic model with spontaneous recurrent seizures" paragraph starting with "While the superiority and novelty of the proposed SNF modality", shown in a new figure (Figure 8).

2. Second, the animals in this study were anesthetized for the duration of the recording. This is confusing, and confounding. They are actually studying the stimulation on the background of ketamine anesthesia, which reduces the clinical and translational relevance.

Response: We appreciate the reviewer for this valuable comment. As pointed in the comment, it is widely known that several anesthetics can affect the neuronal responses in the brain, such as inhibitory effects of seizure induction. Among them, ketamine is an NMDA channel blocker and thus may delay the initiation of seizures or reduce the severity of neuronal loss induced by kainic acid (KA)^{1,2}. Nevertheless, ketamine has been widely used in acute studies with related topics as it can reduce sudden death in animals^{3,4}. Furthermore, in our preliminary tests, we have confirmed that broadly-applied currents (WF stimulation) to the hippocampal region induced spreading fields and thus can elicit excessive motor and sensory responses in an awake rat model. Although this phenomenon was not observed during the proposed SNF stimulation due to the fact that it localizes the electric field well on the target site while the target therapeutic effect is still achieved, anesthesia was still required for investigating acute WF stimulation under which the animal may suffer from the undesired motor and sensory responses. In this reason, we had to use ketamine as an anesthetic in this study. In addition, with new experimental results using a chronic awake model with SRSs, we believe that the clinical and translational relevance of SNF stimulation was adequately validated.

3. Third, the representative seizures they show in Fig 2 suggest that their detection algorithm is sluggish. They are stimulating near the very tail end of seizures to begin with, making their effects less robust.

Response: We really appreciate the reviewer for this important comment. We believe that this comment arises from the misunderstanding caused by the ambiguity of our writing in the previous manuscript. To determine the seizure-suppressing effect of overall hippocampal stimulation, pre-programmed stimulus was repetitively injected in experiments as shown in Figure 2. In other words, the stimulation timing was assigned randomly without any trigger of the detection algorithm since this stimulation scheme was meant to represent the open-loop stimulation. Then, to show the timely seizure detection of our algorithm, we indicated the detection timing of our algorithm

by configuring the figure with green boxes representing the detection zone. As shown in the updated figure, the electrographic seizure was detected timely by our developed detection algorithm confirming that there was no notable lag in the algorithm. The representative result (Figure 2c and e in the previous manuscript) was adopted because we intended to present both growing and suppressing electrographic seizures within one figure. However, in reviewing this valuable comment, we realized that our data and writing might mislead the readers to understand such that our detection algorithm had bad performance due to the long non-response time from seizure generation to stimulation point. Thus, we added a new representative result of rapid inhibition in Figure 2 and modified text to further explain the experimental objective and result, in the Results section of the revised manuscript; “Anti-seizure effects of hippocampal intervention” paragraph starting with “Especially, stimulation succeeded in suppressing seizures”. In conclusion, we believe that the new figure demonstrates excellent performance of our algorithm in detecting the electrographic seizure without notable lag.

4. Fourth, the results in Fig 3D are not very robust. They analyze each bin with an independent Mann Whitney U test. A repeated measures ANOVA would be more appropriate. The independent values are not shown, making it impossible to really understand what the distribution is.

Response: We appreciate the reviewer for this comment regarding the statistical analysis. As recommended, we conducted a repeated measures ANOVA to analyze the termination probability during unilateral and bilateral wide-field stimulation. In addition, the figure (Fig. 3d in the previous manuscript) was modified that the raw data were shown for a better understanding of the data distribution (Fig. 3f in the revised manuscript).

5. Fifth, there is basically no benefit to the sequential narrow field stimulation (SNF) over the wide-field stimulation in terms of stimulation efficacy, and the SNF stimulation was the major point of novelty for the manuscript.

Response: We believe that this comment brings the most critical point about the study. With this comment, we realized that our experiment might not be properly designed to show the intended contribution of our proposed method. In our original experimental design, we thought that it was important to show that the SNF stimulation could modulate the entire hippocampus just like the WF stimulation. However, with this comment, we recognized that the main focus should be set on the SNF stimulation to clearly demonstrate its superiority and novelty. In the previous manuscript, the stimulus intensity of both SNF and WF stimulation was determined via intensity titration until reliable anti-seizure effects were observed. However, we also confirmed that there was a significant difference in electric fields spreading out of the hippocampus during SNF and WF stimulation via measurements *in vivo* and *in silico* (Fig. 4a–h; Supplementary Fig. 4). From these results, it is now rational to establish a different safety level of stimulus intensity for SNF and WF stimulation and use those stimulus intensities in experiments to appropriately compare the therapeutic effects. In this context, we decided to redesign the experimental protocol that compares the seizure-inhibition effect of SNF and WF stimulation with focusing on the superiority of SNF stimulation over WF stimulation via adopting the concept of safety threshold of stimulus intensity defined as a maximum current level that does not cause excessive motor and sensory responses. Since it has been reported that high-frequency (50–200 Hz) pulse stimulation could suppress epileptic activity in hippocampal slices with a field intensity of 50–250 mV/mm^{5,6}, we defined that electric fields of at least 100 mV/mm should be induced in the stimulation target, hippocampus, and off-target spreading fields should be formed as small as possible to exclude unwanted tissue activation. We also experimentally found that WF stimulation elicited abnormal motor responses at much lower intensities compared to SNF stimulation in an awake rat model (Fig. 4l), and this result corresponded with the neuronal responses estimated via computational methods (Fig. 4k). With the experimental results from measuring the induced field distribution and motor response threshold during SNF and WF stimulation as shown in Figure 4, we were able to establish the maximum intensity in the safety range of SNF and WF stimulation (650 and 140 μ A for each condition), satisfying the above conditions (maximum intensity that does not induce off-target tissue activation), and then conducted experiments again to confirm the effects of both stimulations. The results showed that, at each maximum safety level, WF stimulation suppressed less epileptic brain activity compared to SNF stimulation, resulting in significant differences in the success rate of seizure termination and total seizure duration (Fig. 5). We have updated these results to Figure 4 and 5 and also modified the paragraph in the Results section of the revised manuscript; “Validation of the SNF stimulation method”

paragraph starting with “The therapeutic efficacy of overall hippocampal stimulation”. In conclusion, SNF stimulation utilizes highly localized electric field within the target site and thus undesired stimulation due to field spreading is very unlikely. However, the individual micro-stimulation for SNF stimulation is not enough to modulate the entire volume of the hippocampus and thus sequential activation of this micro volume is important for anti-seizure effect. In other words, unorganized micro-stimulation wouldn't suppress the seizure as needed (Supplementary Fig. 6). In addition, each activation channel to induce multiple localized fields can be overlapped freely in sequence, thus the field can be formed much further precisely over conventional methods. Therefore, sequentially ordered micro-stimulation is fundamentally the novelty and superiority of the proposed SNF stimulation for alleviating seizures in temporal lobe epilepsy.

Overall the manuscript presents some interesting data, but is limited in its novelty and represents a somewhat incremental advance to the field of neurostimulation for epilepsy. I would expect to see this in a more specialized journal (e.g., Brain Stimulation), as I do not think this is of interest to a broad audience.

Response: With our new findings and revisions, we believe that the novelty and superiority of the proposed SNF stimulation against the existing hippocampal stimulation techniques are clearly presented in the new manuscript. Also, closed-loop technology for epilepsy is a very important topic in the scientific community and thus we believe that our manuscript will be of interest to the readership of *Nature Communications*.

Reviewer 2:

This manuscript reported a sequential narrow-field (SNF) stimulation method by applying sequential electrical stimulation on pairs of electrodes implanted in the hippocampus to control the localization of electric field. The authors compared the effects of open loop DBS, unilateral and bilateral closed loop wild field stimulation and SNF stimulation on the inhibition of seizure activities on an acute KA seizure model. The results showed the SNF stimulation method could achieve similar control effect similar as bilateral wild field DBS, but with less volume of infected brain area. This research is generally well designed and the manuscript is well organized. However, the idea of using multi implanted electrode to delivery small electric current to control the activation volume is lack of novelty, even this method is designed for seizure control. There are also several major questions need to be resolved:

Response: We appreciate the reviewer for the detailed comments on our manuscript. Reviewing this comment, we realized that our experimental strategy and writing in the previous manuscript for presenting the benefit and superiority of SNF stimulation could lead the readers to misunderstand that our proposed method lacks novelty. However, we would like to take this opportunity to emphasize that our proposed SNF method is fundamentally different from a conventional multi-channel stimulation method that simply activates multiple electrodes. In SNF stimulation, multiple localized fields are induced sequentially to synchronously modulate the neuronal network thus each activation channel can be overlapped freely in sequence to derive the desired field distribution, resulting in the fields can be formed much further precisely in terms of spatial resolution, compared to conventional methods. To the best of our knowledge, there has been no deep brain stimulation method proposed to control brain rhythms elaborately like the SNF modality. In addition, we believe that the concept of complete spatiotemporal control of entire hippocampal region, the origin of most temporal lobe epilepsy, is highly new and innovative. In summary, we are not proposing the electrode fabrication but the stimulation modality, and importantly, our proposed modality can be adopted to the existing directional deep brain leads making it highly practical for clinical translation. Detailed rationale to support our arguments are provided in the responses below and in the text of the revised manuscript.

1. the authors mentioned the SNF stimulation method could modulate the entire hippocampal region. However, the activate volume of electric current is very complex, not only depend on the parameters of electrical stimulation and the electrode shape, but also depend on the physiological characteristic of the targeted brain regions. There is not enough evidence to prove the modulation of entire hippocampus with the SNF method by four pairs of parallel placed electrodes.

Response: We appreciate the reviewer for this important comment regarding the complexity of the volume of activation due to electric current in living tissues. As pointed in the comment, the volume of tissue activated by external electrical stimulation is mainly determined by the induced electric field distribution and the neuronal excitability of the cell. In this context, since the SNF stimulation for alleviating temporal lobe epilepsy targets the hippocampus, mostly comprised of neuronal cells which are similar in the electrical properties, and thus homogeneous consideration would be a valid assumption for electromagnetic computation (also widely accepted and used in the scientific community). In addition, there would be little difference in the physiological properties of the tissue through which the stimulus current passes. To address this concern in unknown worst cases, however, we also investigated the induced field distribution when a tissue had heterogeneous electrical properties compared to that of homogeneous condition using the electromagnetic (EM) computational method. The results showed that there were no significant differences in the induced gradients between heterogeneous and homogeneous conditions (Supplementary Fig. 7). In addition, we had already performed EM simulations with varying the position and number of electrodes when designing and optimizing the electrode montage for the SNF stimulation that can modulate the entire hippocampus while minimizing spreading fields. However, with this comment, we realized that omitting the SNF modality design procedure resulted in a lack of evidence to support our proposed method. Thus, we have added a new figure showing the simulation results for electrode array design (Supplementary Fig. 1), and also added a paragraph explaining the evidence of design and optimization, to the Methods section of the revised manuscript; "Induced electric field design for SNF stimulation". In conclusion, we demonstrate thorough extensive simulation studies to verify that the proposed SNF stimulation is an adequate modality to modulate the entire hippocampus while minimizing the field spreading issue and the four pairs of electrodes (not shape) are

needed to bring anti-seizure effect in the rat model while the number of electrodes may be different for human brain.

2. There might be other related brain nucleus activated by electrical stimulation in hippocampus, since the hippocampus has many adjacent structures and the current could preferentially activate the passing axons from those adjacent structures. What's the effect of SNF stimulation of hippocampus on these adjacent structures in not well illustrated and explained.

Response: We appreciate the reviewer for this important comment. As the reviewer pointed out, the hippocampus has many adjacent structures, including thalamus, entorhinal cortex, parahippocampal gyrus, and amygdala. In addition, the hippocampus and these structures are not only located close together, but also connected to each other via nerve axons. In this context, SNF stimulation has the advantage in precisely delivering stimulation to the target tissue, the hippocampus, while preventing the spreading fields that could directly activate adjacent structures. Therefore, we would like to emphasize that the primary benefit of the proposed SNF stimulation is to reduce the unwanted activation of the neighboring structures as the reviewer commented. In addition, as in the answer to the 4th question, additional experiments for cognitive studies in an awake rat model showed no cognitive-behavioral changes in stimulated rats compared to non-stimulated rats, demonstrating the safety with regional specificity of SNF modality (Fig. 8). However, for more reliable clinical translation of SNF stimulation, it might be needed to further investigate physiological responses via human studies due to the anatomical differences between humans and rodents. As the reviewer recommended, we added these considerations for possible adverse effects of indirect tissue activation to the Discussion section of the revised manuscript; paragraph starting with "In the rodent brain, the cerebral cortex".

3. In the experiment of comparison of firing field effects of SNF stimulation (Fig 5). The author measured the induced intracerebral electric fields on the contralateral motor cortex. Why not check the ipsilateral cortex and other related brain regions, such as those important area in papez circuit?

Response: In the previous manuscript, we believed that it was sufficient to determine the spreading fields by checking their linear characteristics. However, as the reviewer pointed out, we found that it would be insufficient to judge the spreading field distribution in the whole brain based on linear characteristics alone, thus we decided to conduct additional animal experiments to measure the induced electric fields throughout the whole brain. The results showed that the spreading field during WF stimulation was 5–10 times greater than that of SNF stimulation. We added these results to Figure 4 and modified the text to describe the data in Figure 4, to the Results section of the revised manuscript; "Validation of the SNF stimulation method" paragraph starting with "The therapeutic efficacy of overall hippocampal stimulation", and added the method for two-dimensional field measurements to the Methods section of the revised manuscript; "Induced electric field measurements" paragraph.

4. An acute status epilepticus (SE) model was used in this study. This model could activate frequent abnormal brain activities, but may not really reflect epilepsy situation of brain. The activation of SE state is not stable with animals and may fade away with short duration. It is better to test on a chronic seizure animal model. By the way, it is hard to do cognitive behavioral test on acute model. Although the authors mentioned less influence of SNF stimulation on normal cognitive behavior of animals, there is no evidence in the manuscript to prove this hypothesis.

Response: In reviewing this important comment, we realized that an experimental validation of the SNF stimulation should be conducted on a chronic seizure model. Thus, additional experiments using a survival model with spontaneous recurrent seizures were performed to raise assurance of the validity of translational relevance for temporal lobe epilepsy. In addition, we investigated the cognitive-behavioral response using an awake rat model, and the results determined that SNF stimulation does not induce notable cognitive-behavioral changes compared to non-stimulated conditions. We added the experimental results in Figure 8, and added a paragraph describing this new finding to the Results section of the revised manuscript; "SNF stimulation in a chronic model with spontaneous recurrent seizures" paragraph starting with "While the superiority and novelty of the proposed

SNF modality” and an experimental protocol to the Methods section of the revised manuscript; “Experimental details for the chronic experiments”.

5. In fig7a, compared with non-sitm group, why the SNF stimulation caused more Nissl positive neurons in CA3 area but have no difference on the other stimulated areas? What’s the effect of bilateral stimulation group?

Response: We believe that this comment arose from the lack of our explanation. KA induces hypersynchronous firing and paroxysmal discharges, resulting in intense depolarization that causes a sustained Ca^{2+} influx. In the acute phase of epilepsy, it has been reported that KA-induced seizures result in neurodegeneration in CA3 of the hippocampus, whereas CA1, CA2 and DG granule cells are relatively resistant to KA-induced seizures⁶. Since our experimental results of Nissl staining were very similar to those reported in previous studies, we judged that these results were valid. However, we realized that our writing in the previous manuscript for explaining the Nissl staining results was insufficient for the readers to fully understand our intention, and thus we modified the text to further explain the data, in the Results section of the revised manuscript; “Biological analysis for the seizure-suppression effects of SNF stimulation” paragraph starting with “We further verified the expression of Nissl bodies and ionized calcium-binding adapter molecule 1 (Iba1)”.

In the previous manuscript, we considered it important to show the therapeutic effect of SNF stimulation compared to sham controls. However, as the reviewer recommended, we decided to conduct additional experiments for histological analysis of WF stimulation group. These new findings were added in Figure 6 and 7, and the text describing the results were added to the Results section of the revised manuscript; “Biological analysis for the seizure-suppression effects of SNF stimulation” paragraph starting with “To characterize the neuromodulatory effects of SNF hippocampal stimulation”.

Reviewer 3:

In this manuscript, Kang et al. investigate closed-loop control of temporal lobe seizures using a sequential narrow-field (SNF) stimulation method. This method is proposed to decrease the potential side effects of stimulation by reducing the region of brain tissue under the influence of the stimulation without compromising seizure termination efficacy. The motivation and concept for the work are interesting, but key points remain unaddressed and I have several questions regarding the analysis. Overall, further work is needed to improve understanding of this stimulation protocol, its efficacy, and potential for therapeutic applications. At present, the results presented are not sufficient to support the conclusions of the authors regarding the utility of this approach.

Response: We appreciate the reviewer for the constructive comment. As the reviewer pointed out, we added and revised significant amount of contents to improve the manuscript by providing rationales supporting our proposed SNF modality. Detailed point-by-point responses are as below:

1. The innovation of the SNF method is purportedly its restricted gradient across adjacent brain structures. The data presented in Figure 5 incompletely characterize the method. Cortical sites in the contralateral hemisphere are presented for assessment of potential. Why were these sites selected? What about the ipsilateral hemisphere, where presumably the gradients are much higher? What about deep structures (such as thalamus or basal ganglia) that are in relatively close proximity to the hippocampus? How did the potential change over time with the sequential stimulation? The electric field is less for the SNF method, but is it still sufficient to drive/entrain action potential firing? Fields of 1mV/mm have been found to affect neural activity, so over what brain area could the SNF stimulation potentially change activity patterns? Motor responses were used to assess effects on these networks, but are these responses mediated primarily by cortex, or deep structures on the motor output pathway (e.g. basal ganglia or white matter pathways)? Overall, more detailed work to demonstrate the fields generated by the SNF and their effect on neural activity outside of the hippocampus are required.

Response: We appreciate the reviewer for this constructive comment and deep understanding of our experimental design. For a more concise and clear answer, we have responded to this comment via dividing question-by-question as below:

- 1) The innovation of the SNF method is purportedly its restricted gradient across adjacent brain structures. The data presented in Figure 5 incompletely characterize the method. Cortical sites in the contralateral hemisphere are presented for assessment of potential. Why were these sites selected? What about the ipsilateral hemisphere, where presumably the gradients are much higher? What about deep structures (such as thalamus or basal ganglia) that are in relatively close proximity to the hippocampus?

Response: In the previous manuscript, we thought that it was enough to define the properties of spreading fields via their linear characteristics measured considering the direction of the main field vectors. However, we realized that judging the spreading field distribution in the whole brain based on linear properties alone lacked evidence to support the claim, and thus we designed and conducted further *in vivo* experiments. Since a rat's brain is very small, the field can be dispersed or messed up by the metal electrodes if electrode arrays for potential measurement are inserted into all regions of interest. Thus, we measured induced voltage gradients at 54 distinct positions all over the brain while minimizing the interaction between the electrode and induced electric fields, and these new results are inserted into Figure 4 in the revised manuscript. The experimental results (Fig. 4b) clearly supported our claim that in WF stimulation, the induced fields broadly spread, where as in SNF stimulation, the gradients were concentrated in the target region (hippocampus). In addition, these findings were further supported by investigating the field distribution in deep structures during stimulation using a computational method (Supplementary Fig. 4). We modified the texts to explain the redesigned experiments and their results in the Results section of the revised manuscript; "Validation of the SNF stimulation method" paragraph starting with "The therapeutic efficacy of overall hippocampal stimulation", and the Methods section of the revised manuscript; "Induced electric field measurements" paragraph.

- 2) How did the potential change over time with the sequential stimulation?

Response: We conducted additional experiments to investigate the transient response of induced potentials during SNF stimulation, and no effect or interference of sequentially induced gradients on each other was

observed (Supplementary Fig. 3). With these results, we expect that the therapeutic effects of SNF modality would be related to neural regulation at the network level or orchestrated ionic changes in the membrane, induced by sequentially injected stimuli. With this additional finding, it should be noted that further studies on the cellular or network level mechanism of SNF stimulation using brain slices should be conducted to confirm the clinical feasibility of SNF stimulation. We added these observations in the Discussion section of the revised manuscript; paragraph starting with “As a potent analog of glutamate”.

- 3) The electric field is less for the SNF method, but is it still sufficient to drive/entrain action potential firing? Fields of 1mV/mm have been found to affect neural activity, so over what brain area could the SNF stimulation potentially change activity patterns?

Response: In our original experimental design, we designed the electric fields in the hippocampal region sufficient to suppress seizures using EM simulations (Supplementary Fig. 1), based on the literature on hippocampal slices^{5,6}. Also, in this study, since pulsed current, not slow AC electric field, was applied for SNF modality, the threshold to fire the neural networks could be much higher than 1 mV/mm^{7,8}. We therefore compared the proportions of the area where the off-target spreading field exceeds 10 mV/mm (widely accepted as necessary to affect the associated network rhythms⁷, Fig. 4d) and 50 mV/mm (reported to be needed to suppress seizures with pulsed stimulation^{5,6}, Fig. 4e). In both comparisons, the results showed significant differences between SNF and WF stimulation, implying that the proposed SNF stimulation prevent undesired direct activation or inhibition of off-target tissues much better than the WF stimulation. Furthermore, these findings were further supported by the electromagnetic simulation results (Supplementary Fig. 4). We have added these findings and considerations to the Results section of the revised manuscript; “Validation of the SNF stimulation method” paragraph starting with “The therapeutic efficacy of overall hippocampal stimulation”, and the Methods section of the revised manuscript; “Induced electric field design for SNF stimulation” paragraph.

- 4) Motor responses were used to assess effects on these networks, but are these responses mediated primarily by cortex, or deep structures on the motor output pathway (e.g. basal ganglia or white matter pathways)? Overall, more detailed work to demonstrate the fields generated by the SNF and their effect on neural activity outside of the hippocampus are required.

Response: We further investigated the electric field distribution using EM simulations to determine whether the motor responses were mediated directly by the cortex, or deep structures on the motor output pathway (Supplementary Fig. 4). The simulation results showed that > 100 mV/mm of electric fields were sophisticatedly concentrated in the hippocampal region in SNF condition, however, the induced fields during WF stimulation broadly spread to the cortex, and basal ganglia including the striatum and thalamus. We already confirmed that excessive motor responses were only observed during WF stimulation via *in vivo* experiments, implying that SNF stimulation does not affect neuronal rhythm outside of the hippocampus. Taken together, these findings suggest that SNF stimulation can induce fields in the hippocampal region sufficient to modulate network rhythms while preventing undesired side effects outside of the hippocampus. We have added these results and further considerations to the Results section of the revised manuscript; “Validation of the SNF stimulation method” paragraph starting with “The therapeutic efficacy of overall hippocampal stimulation”, and the Discussion section of the revised manuscript; paragraph starting with “With *in vivo* and *in silico* approaches”.

2. The mechanism of the SNF in seizure termination is not directly investigated. This is a major gap in the present work. How many sites are required for this method? The authors use 4, but what determines this amount? What is the appropriate spacing – minimum and maximum? What about the orientation? Is it important that the sites activate sequentially in a particular order? The authors seem to orient the electrodes along the dorsoventral axis, perhaps in CA1 at each location, but this is not specified. Do the electrodes need to be activated from dorsal to ventral or vice versa, or is it directionally independent? The authors do not seem to survey ventral hippocampus. Is the seizure terminated throughout the hippocampus by the regional stimulation that they apply? Without this fundamental understanding of the method, it is difficult to appreciate its applicability and translational potential.

Response: We appreciate the reviewer for this valuable comment regarding the mechanistic investigation of our proposed SNF stimulation. For a more concise and clear response, we have answered this comment via dividing question-by-question as below:

- 1) The mechanism of the SNF in seizure termination is not directly investigated. This is a major gap in the present work. How many sites are required for this method? The authors use 4, but what determines this amount? What is the appropriate spacing – minimum and maximum? What about the orientation?

Response: Electrode arrays for the proposed SNF stimulation were designed via literature surveys, and our preliminary *in vivo* and *in silico* works. From the literature survey, we pre-defined that the stimulation pulse needs to induce an electric field of at least 100 mV/mm in the entire hippocampal region to suppress epileptic rhythms^{5,6}, while off-target fringing fields should be formed as small as possible to minimize the undesired neuronal activation. To determine an optimized electrode montage that can minimize tissue damage and maximize therapeutic effect, we investigated the induced field distribution during SNF stimulation with varying the number and position of electrodes, and stimulus intensity using a computational method (Supplementary Fig. 1). Since it is widely known that field direction is important only for DC stimulation and not for AC or pulsed stimulation in seizure suppression stimulation⁵, no consideration of field orientation for each electrode is given here and it is more likely to be feasible in an *in vivo* condition in which the direction of the axon is highly diverse. With the simulation results, we found that as the number of electrodes increased, the stimulus intensity required to inhibit seizures decreased, while the absolute value of the decreasing slope gradually decreased (Supplementary Fig. 1b). Also, it was confirmed that the off-target spreading fields rapidly decreased with the increase in the number of electrodes in SNF stimulation, but this effect was finally plateaued when there were more than 4-pairs (Supplementary Fig. 1c) and additional electrodes were not justified anymore. Thus, we decided to adopt a 4-pairs configuration that relatively less causes tissue damage due to electrode insertion and can induce a field sufficient to suppress epileptic networks while minimizing off-target fringing fields, with relatively low stimulus intensity. We added the design procedure and considerations to the Methods section of the revised manuscript; “Induced electric field design for SNF stimulation” paragraph.

- 2) Is it important that the sites activate sequentially in a particular order?

Response: To address this comment about the importance of fields induced in a particular order, we conducted additional experiments for investigating the seizure-suppressing effect of random-order narrow-field (RNF) stimulation. Then, we confirmed that the seizure-suppressing effect of RNF stimulation was significantly lower than that of SNF stimulation (Supplementary Fig. 6). With these results, we confirmed that the spatiotemporally organized electric field has a superior seizure-inhibitory effect compared to the randomly applied micro-stimulation, and thus we believe that further studies on the cellular or network level mechanisms of SNF stimulation are needed to confirm the clinical feasibility of SNF stimulation. We added texts to discuss these observations and considerations to the Discussion section of the revised manuscript; paragraph starting with “As a potent analog of glutamate”.

- 3) The authors seem to orient the electrodes along the dorsoventral axis, perhaps in CA1 at each location, but this is not specified. Do the electrodes need to be activated from dorsal to ventral or vice versa, or is it directionally independent? The authors do not seem to survey ventral hippocampus. Is the seizure terminated throughout the hippocampus by the regional stimulation that they apply? Without this fundamental understanding of the method, it is difficult to appreciate its applicability and translational potential.

Response: It has been reported that the injections of KA were found to induce behavioral seizures and neurodegeneration in the dorsal hippocampus in a rodent model^{9,10}. And in our pilot test to confirm the feasibility of overall hippocampal stimulation, we confirmed that delivering stimulation to the ventral hippocampus did not significantly show the seizure-inhibitory effects compared to those of the dorsal hippocampus. Also, it was confirmed that the direction of electrode activation, from dorsal to ventral or vice versa, did not significantly affect the therapeutic effects, and we believe that this result was due to the fact that the seizure-suppressing effect by pulsed stimulation is not dependent on the orientation between the fields and neurons⁵. Thus, the experimental protocol for this study was designed to modulate the dorsal hippocampal region with the stimuli from ventral to dorsal direction. Although the experimental results showed the superior therapeutic effect via dorsal hippocampal control in a rodent model, since the brain structures of humans and rodents are quite different, we believe that further investigations are necessary by considering the differences between them for clinical translation while the benefit of the proposed SNF stimulation is expected to be similar. We added texts to discuss these observations and considerations to the Methods section of the revised manuscript; “Induced electric field design for SNF stimulation” paragraph starting with “Next, we pre-defined that the stimulation pulse”.

3. It seems as though there is a substantial lag between seizure onset and delivery of stimulation (see Figure 1). In the example provided, the stimulation (orange) occurs at least > 15 s after the onset of the seizure. Was this duration standardized for all the seizures? Time prior to stimulation initiation could be a major confounder to overall seizure duration, as once the seizure has been established for longer it could be harder to stop.

Response: We believe that this comment also arises from misunderstanding caused by the ambiguity in our explanation. We also agree with the reviewer that rapid on-demand intervention could be a key factor to inhibit seizures. We determined the potential for seizure-suppressing effect of overall hippocampal stimulation with the pre-programmed stimulus (open-loop scheme) that was injected randomly without any trigger of the detection algorithm, and regardless of the stimulation timing, all the seizures were detected rapidly and timely by the detection algorithm and they were indicated with green boxes in Figure 2. Additionally, the detection algorithm was directly applied to the experiments with a closed-loop scheme for rapid and timely interventions as shown in Figure 3, 5, and 8. We realized the readers could misunderstand as the representative result of pre-programmed stimulation seems like having a substantial lag due to the long non-response time to the stimulation-trigger point and the lack of detailed explanation. Thus, we added a result of rapid intervention to Figure 2 to clear up the possible misunderstanding, and modified text to account for new data in the Results section of the revised manuscript; “Anti-seizure effects of hippocampal intervention” paragraph starting with “Especially, stimulation succeeded in suppressing seizures”.

4. The rationale for the histologic experiments is not clear to me. The SNF tissue is compared with unstimulated tissue, but the more apt comparison would be between the wide field (WF) and SNF stimulated tissue to determine whether the SNF tissue is less affected.

Response: As recommended, we conducted additional histologic experiments of WF stimulation group. We added the new data to Figure 6 and 7, and modified the text to explain the data in the Results section of the revised manuscript; “Biological analysis for the seizure-suppression effects of SNF stimulation” paragraph starting with “To characterize the neuromodulatory effects of SNF hippocampal stimulation”, and the Discussion section of the revised manuscript; paragraph starting with “As a potent analog of glutamate”.

5. The methods for calculating the degree of synchronization, and what exactly this means, are not clear. It would be more accurate to compare individual seizure epochs to see whether seizures were expressed bilaterally and quantify this. The authors could separately look at synchronization of other hippocampal rhythms, such as theta, if of interest.

Response: In reviewing this valuable comment, we realized that our data and text might be confusing for readers. Phase synchronization is a widely used method to compare two or more continuous time series of brain activity and that is only sensitive to the phases, irrespective of the amplitudes of the signals^{11,12}. And this method has already been used in lots of studies for epileptic seizure detection and prediction. However, we recognized that it may be difficult to understand for the readers unfamiliar with this method, thus we added Figure 3b and c and explanation for them to the revised manuscript to make it easier to understand the definition and calculation of the degree of synchronization; the Results section of the revised manuscript; “Comparison of neuronal desynchronization by bilateral versus unilateral stimulation” paragraph starting with “Epileptic seizures in patients with TLE can originate from the unilateral or bilateral hippocampus”, and the Methods section of the revised manuscript; “Recording and processing the electrophysiological activity for electrographic seizure detection” paragraph.

As recommended, we conducted additional data analysis for more accurate comparison of individual seizures during unilateral and bilateral WF stimulation. We added the results to Supplementary Figure 2 and modified the paragraph to explain the newly added data, in the Results section of the revised manuscript; “Comparison of neuronal desynchronization by bilateral versus unilateral stimulation” paragraph. We believe that these new findings and analyses show the novelty and benefit of the SNF stimulation for anti-seizure effects.

References

- 1 Bielefeld, P. *et al.* A standardized protocol for stereotaxic intrahippocampal administration of kainic acid combined with electroencephalographic seizure monitoring in mice. *Frontiers in neuroscience* **11**, 160 (2017).
- 2 Lees, G. Effects of anaesthetics, anticonvulsants and glutamate antagonists on kainic acid-induced local and distal neuronal loss. *Journal of the neurological sciences* **108**, 221-228 (1992).
- 3 Choi, H. J., Lee, A. J., Kang, K. S., Song, J. H. & Zhu, B. T. 4-hydroxyestrone, an endogenous estrogen metabolite, can strongly protect neuronal cells against oxidative damage. *Scientific reports* **10**, 1-15 (2020).
- 4 Lee, J. M. *et al.* Morin prevents granule cell dispersion and neurotoxicity via suppression of mTORC1 in a kainic acid-induced seizure model. *Experimental Neurobiology* **27**, 226 (2018).
- 5 Durand, D. M. & Bikson, M. Suppression and control of epileptiform activity by electrical stimulation: a review. *Proceedings of the IEEE* **89**, 1065-1082 (2001).
- 6 Spigolon, G., Veronesi, C., Bonny, C. & Vercelli, A. c-Jun N-terminal kinase signaling pathway in excitotoxic cell death following kainic acid-induced status epilepticus. *European Journal of Neuroscience* **31**, 1261-1272 (2010).
- 7 Vöröslakos, M. *et al.* Direct effects of transcranial electric stimulation on brain circuits in rats and humans. *Nature communications* **9**, 1-17 (2018).
- 8 Neudorfer, C. *et al.* Kilohertz-frequency stimulation of the nervous system: A review of underlying mechanisms. *Brain stimulation* **14**, 513-530 (2021).
- 9 Kienzler-Norwood, F. *et al.* A novel animal model of acquired human temporal lobe epilepsy based on the simultaneous administration of kainic acid and lorazepam. *Epilepsia* **58**, 222-230 (2017).
- 10 Zeidler, Z. *et al.* Targeting the mouse ventral hippocampus in the intrahippocampal kainic acid model of temporal lobe epilepsy. *Eneuro* **5** (2018).
- 11 Salam, M. T., Kassiri, H., Genov, R. & Perez Velazquez, J. L. Rapid brief feedback intracerebral stimulation based on real-time desynchronization detection preceding seizures stops the generation of convulsive paroxysms. *Epilepsia* **56**, 1227-1238 (2015).
- 12 Cobb, S., Buhl, E., Halasy, K., Paulsen, O. & Somogyi, P. Synchronization of neuronal activity in hippocampus by individual GABAergic interneurons. *Nature* **378**, 75-78 (1995).

Reviewers' comments:

Reviewer #1 (Remarks to the Author):

The authors did a commendable job addressing my prior concerns. The inclusion of a model with SRS (chronic KA model) is a substantial improvement in both the impact and the interpretability of the study. Similarly, the additional experiments and analyses now demonstrate superiority of SNF stimulation over WF stimulation, which was really essential for the main focus of the paper. Supporting this, they appropriately updated the statistics, which are now both more appropriate and robust.

While the level of focus may still be a bit narrow for this journal, I have no remaining concerns regarding the methods, approach, or conclusions.

Reviewer #2 (Remarks to the Author):

The authors add a series of experiments and analyses, making the manuscript more clear and complete. Most of the questions I mentioned have been clearly answered. Several minor issues should be considered before the final publication.

1. KA-induced SE model was majorly used in this study. I don't fully agree that this model reproduced nrtopathological and electroencephalographic features in human TLE patient, especially with the acute SE state of this model. I suggest the authors to specify the difference between rodent model and human patient, to make the results of the article more easily accepted by different readers.
2. at the end of Page 9, the authors mentioned " it resulted in a significant decrease in band power during SNF stimulation compared to WF stimulation (Supplementary Fig. 2c)." However, I didn't find the clear label of SNF stimulation in supplementary Fig. 2c. It might be a mistake, please check it again.
3. At the middle of Page 10, "Stimulation currents were sequentially applied to the five electrode configurations (Fig. 4a, bottom right)". Here, "five electrode configurations" should be four electrodes?
4. at the beginning of Page 14, the author mentioned sham group. It is better to uniform the group name in the whole manuscript.
5. the procedure of chronic seizure induction with KA injection is weird, since the author didn't terminate the SE state with diazepam. can authors provide more details or evidence of the success rate of this chronic model.
6. it is not very clear of the method for seizure detection, either with acute SE state or with chronic seizure model. How did the author figure out the false detection? since it is very important feature to be considered in a closed-loop stimulation system, as listed in the following papers:
(1) An Energy Efficient AdaBoost Cascade Method for Long-term Seizure Detection in Portable Neurostimulators, IEEE Transactions on Neural Systems and Rehabilitation Engineering, 27(11), 2019/11
(2) Acute Seizure Control Efficacy of Multi-Site Closed-Loop Stimulation in a Temporal Lobe Seizure Model[J]. IEEE Transactions on Neural Systems and Rehabilitation Engineering, 2019, 27(3):419-428.

Reviewer #3 (Remarks to the Author):

The authors have embarked upon substantial further experimentation and analysis to address the initial review points. However, the manuscript overall continues to suffer from issues related to clarity, methodological detail, and controls.

The authors state in the introduction that off target effects may induce cognitive or affective symptoms when the hippocampus is targeted. It is important to note that affecting just the hippocampus itself could reasonably be expected to result in such symptoms given its role in mediating declarative and emotional memory.

In the “Comparison of neuronal desynchronization by bilateral versus unilateral stimulation” paragraph, the comparison is initially between WF unilateral and bilateral stimulation. Although I understand the rationale that bilateral vs. unilateral stimulation has rarely been investigated in this manner, the results do seem to be quite a deviation from the main hypothesis of the paper which is regarding SNF stimulation. There is a fragment of a sentence in this paragraph commenting on SNF (“...and it resulted in a significant decrease in band power during SNF stimulation compared to WF stimulation (Supplementary Fig. 2c)”) but the interpretation of this comment is unclear. It would be more relevant to discuss the effects of unilateral vs. bilateral SNF stimulation as a main discussion point in the main figure, because conceivably the SNF could be even less efficacious in unilateral configuration than WF. The comparison between unilateral and bilateral WF would then be supplementary to, and supportive of, this point.

Regarding Figure 5, there is confusion about the anatomic range of effects for the SNF stimulation. If the effects of SNF are restricted to the hippocampus at field strengths used for seizure termination, then why does SNF significantly affect cortical rhythms in Figure 5? If this is related to termination of a hippocampal seizure that has already propagated to cortex, such that isolated hippocampal stimulation terminates it, this would have to be demonstrated. As currently presented, the analysis performed is not clear.

Regarding Figure 6, what is the timing for sacrifice of the animals after WF/SNF stimulation? How was the amount of stimulation delivered controlled if it was delivered in a closed-loop fashion, driven by spontaneous occurrence of events in any given animal? Why does the WF stimulation reduce cFos activation only in cortex? Why is cFos activation higher in DG for the SNF stimulation condition? It is stated that number of GAD65 neurons was quantified to “verify the contribution of GABAergic neurons to the seizure suppressing effects observed during stimulation.” Identifying just the number of GAD65 positive neurons does not allow interpretation about functional properties during a stimulation epoch.

Although including the chronic seizure model is laudable, again, the comparison here should be between WF and SNF stimulation rather than just SNF and non-stimulated animals. Furthermore, the behavioral tests are only performed during SNF stimulation; would WF stimulation, which causes off target effects, cause detrimental effects on these tasks? If not, the tasks may not be sensitive enough to capture such effects.

In the “Anti-seizure effects of hippocampal intervention” paragraph, it would be important to specify the mean and range of latencies at which seizures are terminated after stimulation.

Figure 2 is focused on WF stimulation in the acute model, and the conclusion is that this stimulation is highly effective in suppressing seizures. Subsequently, this same WF stimulation is shown to have a seizure termination rate of only 0.57. The organization and presentation of the data make the manuscript extremely difficult to follow.

In the introduction, please note that it should be “temporally” and not “temporarily.”

Responses to the reviewers' comments

We appreciate the reviewers for their constructive feedbacks on our first revised manuscript. All the comments were very helpful in improving our manuscript, and we have addressed them via additional analyses and experiments in the revised manuscript. Our major rework in this revision is summarized below:

- 1) We further analyzed the experimental data to clearly demonstrate that the proposed SNF stimulation is capable of termination of fully diffuse seizures as well as early termination of hippocampal seizures.
- 2) To address the reviewer's concern about a deviation of experimental results from the main topic, we conducted additional acute experiments to compare the effects of unilateral versus bilateral SNF stimulation.
- 3) To comprehensively compare the seizure-suppressing effects and cognitive-behavioral responses in non-stimulation, WF stimulation, and SNF stimulation groups, we conducted additional *in vivo* experiments with WF stimulation.

With our additional findings and rationale summarized in this report, we believe that the reviewers' concerns have been clearly addressed. Detailed point-by-point responses to the comments and corresponding changes are presented below.

Reviewers' comments

Reviewer 1:

The authors did a commendable job addressing my prior concerns. The inclusion of a model with SRS (chronic KA model) is a substantial improvement in both the impact and the interpretability of the study. Similarly, the additional experiments and analyses now demonstrate superiority of SNF stimulation over WF stimulation, which was really essential for the main focus of the paper. Supporting this, they appropriately updated the statistics, which are now both more appropriate and robust.

While the level of focus may still be a bit narrow for this journal, I have no remaining concerns regarding the methods, approach, or conclusions.

Response: We highly appreciate the reviewer for the constructive feedbacks regarding our additional experiments and revised manuscript to improve the original manuscript based on the referee's comments in the first round. As commented by the reviewer, substantial improvements could be made via additional analyses with appropriate statistical methods and experiments with a SRS model suggested by the referee.

Reviewer 2:

The authors add a series of experiments and analyses, making the manuscript more clear and complete. Most of the questions I mentioned have been clearly answered. Several minor issues should be considered before the final publication.

Response: We appreciate the reviewer for agreeing with our extensive efforts on additional experiments and analyses to improve the clarity and completeness of the original manuscript. We happily address additional issues pointed out by the reviewer here, and detailed point-by-point responses are as below:

1. KA-induced SE model was majorly used in this study. I don't fully agree that this model reproduced neuropathological and electroencephalographic features in human TLE patient, especially with the acute SE state of this model. I suggest the authors to specificate the difference between rodents model and human patient, to make the results of the article more easily accepted by different readers.

Response: We appreciate the reviewer for this valuable suggestion to discuss the translational relevance of the animal model due to the imperfect match between KA-induced models and human TLE patients. As the reviewer pointed out, especially, the acute SE model may present symptomatic features somewhat different from those of human TLE patients due to the short duration and excessive recurrence. Thus, further experiments and analyses had been conducted using the chronic model to address these problems in the first round, but with this comment, we recognized that the chronic model also may not fully reproduce neuropathological and electroencephalographic features in human TLE patients. Therefore, we added text to discuss these considerations to the Discussion section of the revised manuscript.

Modifications:

Page 21, line 475–482: “In the present study, acute and chronic seizure rodent models were used ... the validity of translational relevance of SNF stimulation for human TLE patients.”

2. at the end of Page 9, the authors mentioned " it resulted in a significant decrease in band power during SNF stimulation compared to WF stimulation (Supplementary Fig. 2c)." However, I didn't find the clear label of SNF stimulation in supplementary Fig2c. It might be a mistake, please check it again.

Response: We appreciate the reviewer for pointing out this mistake. We carefully corrected all the typos and errors throughout the manuscript again, and the detailed label in Supplementary Fig. 2c in the Results section was updated correctly this time.

Modifications:

Page 9, line 192–197: “As a result, the total seizure duration of bilateral stimulation was 18.6% less than that of unilateral one (Fig. 3g; 20.3% and 38.9% for bilateral and unilateral WF stimulation, respectively; Mann–Whitney U test; $P = 0.0226$ for comparison between the two conditions; and $P = 0.9362$ for all in-group comparisons between left and right hemisphere), and it resulted in a significant decrease in band power during bilateral WF stimulation compared to unilateral stimulation (Supplementary Fig. 2c).”

3. At the middle of Page 10, "Stimulation currents were sequentially applied to the five electrode configurations (Fig. 4a, bottom right)". Here, "five electrode configurations" should be four electrodes?

Response: We believe that this comment arose from the misunderstanding caused by ambiguity in our writing. To meticulously investigate the induced intracerebral electric fields during WF and SNF stimulation, we individually delivered stimulation currents for each channel (WF, R1, R2, R3, and R4 configurations as shown in Fig. 4a) during potential recording. In addition, it was confirmed that there was no effect or interference of sequentially induced fields on each other during SNF stimulation (Supplementary Fig. 3). We modified the text to clarify the stimulation protocol for field measurements in the Results section of the revised manuscript as below.

Modifications:

Page 10, line 214–217: “Stimulation currents were individually applied to each five electrode configurations (Fig. 4a, bottom right), and the intracerebral potential was recorded for calculating the induced voltage gradients (Fig. 4b; Supplementary Fig. 3).”

4. at the beginning of Page 14, the author mentioned sham group. It is better to uniform the group name in the whole manuscript.

Response: We highly appreciate the reviewer for this feedback. We believe that the recommendation of unifying the group name throughout the manuscript will substantially improve the readability of the manuscript. As recommended, each group name was unified to clarify the distinction between the experimental groups in the revised manuscript as below.

- 1) Wide-field stimulation (WF-stim; no safety limit): used in Fig. 3, 4, and 8e-i.
- 2) Low-intensity wide-field stimulation (LIWF-stim; WF-stim with a safety limit): used in Fig. 5, 6, 7, and 8c, d.
- 3) Sequential narrow-field stimulation (SNF-stim): used in Fig. 4, 5, 6, 7, and 8.
- 4) Non-stimulation (Non-stim; control group with KA injection and no treatment): used in Fig. 5, 6, 7, and 8.

5. the procedure of chronic seizure induction with KA injection is weird, since the author didn't terminate the SE state with diazepam. can authors provide more details or evidence of the success rate of this chronic model.

Response: The chronic kainic acid (KA) model without diazepam injection for terminating status epilepticus (SE) was investigated and demonstrated in many previous studies¹⁻⁴. In these studies, multiple low-dose KA injections method was used to minimize the mortality rate that is typically associated with a single high-dose injection. In addition, many previous studies reported that more than 90% of kainate-treated rats, which experienced motor seizures (Racine stages III–V) in SE and survived after SE, developed spontaneous recurrent seizures. To better explain this, we added the references to the Methods section of the revised manuscript.

Modifications:

Page 25,

Ref 95: Hellier, JL., et al. "Recurrent spontaneous motor seizures after repeated low-dose systemic treatment with kainate: assessment of a rat model of temporal lobe epilepsy." *Epilepsy research* 31(1), 73-84. (1998)

Ref 96: Williams, PA., et al. "Development of spontaneous recurrent seizures after kainate-induced status epilepticus." *Journal of Neuroscience* 29(7), 2103-2112. (2009)

Ref 97: Rao, MS., et al. "Hippocampal neurodegeneration, spontaneous seizures, and mossy fiber sprouting in the F344 rat model of temporal lobe epilepsy." *Journal of neuroscience research* 83(6), 1088-1105. (2006)

Ref 98: Rao, MS., et al. "Strategies for promoting anti-seizure effects of hippocampal fetal cells grafted into the hippocampus of rats exhibiting chronic temporal lobe epilepsy." *Neurobiology of disease* 27(2), 117-132. (2007)

6. it is not very clear of the method for seizure detection, either with acute SE state or with chronic seizure model. How did the author figure out the false detection? since it is very important feature to be considered in a closed-loop stimulation system, as listed in the following papers:

- (1) An Energy Efficient AdaBoost Cascade Method for Long-term Seizure Detection in Portable Neurostimulators, *IEEE Transactions on Neural Systems and Rehabilitation Engineering*, 27(11), 2019/11
- (2) Acute Seizure Control Efficacy of Multi-Site Closed-Loop Stimulation in a Temporal Lobe Seizure Model[J].

IEEE Transactions on Neural Systems and Rehabilitation Engineering, 2019, 27(3):419-428.

Response: We fully agree with the reviewer's comment on the importance of detection algorithms in a closed-loop stimulation scheme. In the acute studies, since the electrodes were firmly fixed to the skull with dental cement and the animals were anesthetized during experiments, our closed-loop stimulation system could continuously record stable local field potentials. In addition, fine-tuning the detection algorithm may not be appropriate for short durations of acute experiments. Thus, electrographic seizures from acute models were detected using the power spectral density alone^{5,6}. In the chronic study, the detection algorithm was constructed based on the power spectral density as in the acute trial, but the spike characteristics were also used for more accurate seizure detection in freely moving rats⁷⁻⁹. In particular, the false alarms were prevented by excluding abnormal patterns such as excessively large signals and specific repetitive features due to external artifacts. We modified text and added references to further explain the seizure detection algorithm in the Methods section of the revised manuscript as below.

Modifications:

Page 25, line 598–610: "In the acute experiments, the baseline ... the detection process to prevent false alarms."

Ref 50: Zheng, Y, et al. "Acute seizure control efficacy of multi-site closed-loop stimulation in a temporal lobe seizure model." IEEE Transactions on Neural Systems and Rehabilitation Engineering 27(3), 419-428. (2019)

Ref 101: Xu, K, et al. "An energy efficient adaboost cascade method for long-term seizure detection in portable neurostimulators." IEEE Transactions on Neural Systems and Rehabilitation Engineering 27(11), 2274-2283. (2019)

Ref 102: Armstrong, C, et al. "Closed-loop optogenetic intervention in mice." Nature protocols 8(8), 1475-1493. (2013)

Reviewer 3:

The authors have embarked upon substantial further experimentation and analysis to address the initial review points. However, the manuscript overall continues to suffer from issues related to clarity, methodological detail, and controls.

Response: We appreciate the reviewer for the positive feedbacks and comments to improve the manuscript. To address the issues pointed out by the reviewer, we have performed additional *in vivo* experiments and modified text to further explain the data. Detailed point-by-point responses are as below:

1. The authors state in the introduction that off target effects may induce cognitive or affective symptoms when the hippocampus is targeted. It is important to note that affecting just the hippocampus itself could reasonably be expected to result in such symptoms given its role in mediating declarative and emotional memory.

Response: We appreciate the reviewer for this valuable comment regarding the potential adverse effects of hippocampal stimulation itself. In reviewing this comment, we realized that since it is widely accepted that the hippocampus is responsible for declarative and emotional memory, we cannot avoid considering that direct hippocampal control may affect these functions. At this moment, we believe that these concerns shall be further clarified through human clinical studies since they are related to human physiology and psychology. Therefore, we agree that this is a limitation of this study and added texts to discuss these considerations to the Discussion section of the revised manuscript.

Modifications:

Page 18, line 411–421: “Hippocampus is widely accepted as responsible for declarative and emotional memory ... for safer clinical translation of SNF methods.”

2. In the “Comparison of neuronal desynchronization by bilateral versus unilateral stimulation” paragraph, the comparison is initially between WF unilateral and bilateral stimulation. Although I understand the rationale that bilateral vs. unilateral stimulation has rarely been investigated in this manner, the results do seem to be quite a deviation from the main hypothesis of the paper which is regarding SNF stimulation. There is a fragment of a sentence in this paragraph commenting on SNF (“...and it resulted in a significant decrease in band power during SNF stimulation compared to WF stimulation (Supplementary Fig. 2c)”) but the interpretation of this comment is unclear. It would be more relevant to discuss the effects of unilateral vs. bilateral SNF stimulation as a main discussion point in the main figure, because conceivably the SNF could be even less efficacious in unilateral configuration than WF. The comparison between unilateral and bilateral WF would then be supplementary to, and supportive of, this point.

Response: We appreciate the reviewer for this important comment. For a more concise and clear answer, we have responded to this comment via dividing question-by-question as below:

- 1) In the “Comparison of neuronal desynchronization by bilateral versus unilateral stimulation” paragraph, the comparison is initially between WF unilateral and bilateral stimulation. Although I understand the rationale that bilateral vs. unilateral stimulation has rarely been investigated in this manner, the results do seem to be quite a deviation from the main hypothesis of the paper which is regarding SNF stimulation.

It would be more relevant to discuss the effects of unilateral vs. bilateral SNF stimulation as a main discussion point in the main figure, because conceivably the SNF could be even less efficacious in unilateral configuration than WF. The comparison between unilateral and bilateral WF would then be supplementary to, and supportive of, this point.

Response: The ultimate goal of this study has been to validate the superiority and novelty of the proposed SNF stimulation in terms of therapeutic effects and safety. To achieve this, we carefully conducted all experiments and analyses over a long period of time in a phased manner. In the pilot stage prior to defining the SNF configuration for enhanced safety, we first confirmed the seizure suppression efficacy of overall hippocampal

control itself, and compared the effects of unilateral and bilateral conditions using a conventional method, WF stimulation. With the experimental results in this stage, we assumed that the bilateral configuration is a proper montage for a better anti-seizure effect on SNF stimulation. Therefore, we organized the original manuscript following this phased approach. However, with this comment, we also recognized that this assumption needs to be proven for SNF stimulation modality for the readership. Thus, additional experiments and analyses for unilateral versus bilateral SNF stimulation were conducted. However, in order not to undermine the logical flow of the original manuscript that was organized following the phase of the study, we added these new findings as supplementary material and modified the main text to further explain the data, in the Results section of the revised manuscript.

Modifications:

Page 13, line 280–283: “In addition, the anti-seizure effects during unilateral versus bilateral SNF stimulation were investigated, and the results demonstrated the superior effects of the bilateral configuration on spectral density, phase synchrony, and seizure duration during control (Supplementary Fig. 7)”

- 2) There is a fragment of a sentence in this paragraph commenting on SNF (“...and it resulted in a significant decrease in band power during SNF stimulation compared to WF stimulation (Supplementary Fig. 2c)”) but the interpretation of this comment is unclear.

Response: We appreciate the reviewer for the careful review of the manuscript. Supplementary Fig. 2c presents the band power during unilateral and bilateral WF stimulation, not SNF and WF stimulation. We carefully corrected all the typos and errors throughout the manuscript again, and the detailed explanation for Supplementary Fig. 2c in the Results section was updated correctly this time in reflecting this comment.

Modifications:

Page 9, line 192–197: “As a result, the total seizure duration of bilateral stimulation was 18.6% less than that of unilateral one (Fig. 3g; 20.3% and 38.9% for bilateral and unilateral WF stimulation, respectively; Mann–Whitney U test; $P = 0.0226$ for comparison between the two conditions; and $P = 0.9362$ for all in-group comparisons between left and right hemisphere), and it resulted in a significant decrease in band power during bilateral WF stimulation compared to unilateral stimulation (Supplementary Fig. 2c).”

3. Regarding Figure 5, there is confusion about the anatomic range of effects for the SNF stimulation. If the effects of SNF are restricted to the hippocampus at field strengths used for seizure termination, then why does SNF significantly affect cortical rhythms in Figure 5? If this is related to termination of a hippocampal seizure that has already propagated to cortex, such that isolated hippocampal stimulation terminates it, this would have to be demonstrated. As currently presented, the analysis performed is not clear.

Response: We really appreciate the reviewer for this valuable comment. As recommended, we further analyzed the experimental data during SNF stimulation to clearly demonstrate the feasibility of early termination of hippocampal seizures as well as termination of fully propagated seizures. The results showed that SNF stimulation had a significant seizure-suppressing effect not only in hippocampal onset but also in diffuse conditions (Supplementary Fig. 6). It could imply that this strong seizure-suppressing effect of hippocampal SNF stimulation, which worked even in seizures propagated to the cortex, might be caused by stimulus energy delivered via an extrahippocampal connecting pathway. We modified text to further explain the newly added data in the Results and Discussion section of the revised manuscript as below.

Modifications:

Page 13, line 276–280: “Experimental data from non-stimulated and SNF-stimulated groups were further analyzed to investigate the seizure-suppressing effects including early termination of hippocampal seizures and termination of fully propagated seizures, and the results showed that SNF stimulation suppressed epileptic rhythms not only in hippocampal onset but also in diffuse conditions (Supplementary Fig. 6).”

Page 20, line 462–474: “Although the field distribution was restricted to the hippocampal region ... to confirm the exact rationale for these phenomena.”

4. Regarding Figure 6, what is the timing for sacrifice of the animals after WF/SNF stimulation? How was the amount of stimulation delivered controlled if it was delivered in a closed-loop fashion, driven by spontaneous occurrence of events in any given animal? Why does the WF stimulation reduce cFos activation only in cortex? Why is cFos activation higher in DG for the SNF stimulation condition? It is stated that number of GAD65 neurons was quantified to “verify the contribution of GABAergic neurons to the seizure suppressing effects observed during stimulation.” Identifying just the number of GAD65 positive neurons does not allow interpretation about functional properties during a stimulation epoch.

Response: We appreciate the reviewer for this constructive comment and deep understanding of our experiments. For a more concise and clear answer, we have responded to this comment via dividing question-by-question as below:

- 1) Regarding Figure 6, what is the timing for sacrifice of the animals after WF/SNF stimulation? How was the amount of stimulation delivered controlled if it was delivered in a closed-loop fashion, driven by spontaneous occurrence of events in any given animal?

Response: The acute experiments were conducted for 3 h after the kainic acid (KA) injection, and then rats were sacrificed to extract brain tissues that were for immunohistochemical analysis. Stimulation was delivered when the detection algorithm embedded in the proposed system identified electrographic seizures and transmitted a trigger signal. The stimulus amplitude was determined as the minimum value indicating seizure-suppressing effects for each animal via intensity titration with steps of 50 μ A. All of the details related to the experimental protocols are described in the Methods section.

- 2) Why does the WF stimulation reduce cFos activation only in cortex? Why is cFos activation higher in DG for the SNF stimulation condition?

Response: It has been reported that stronger c-Fos expression in dentate gyrus (DG) than other regions is a common feature during the acute phase of epilepsy and the potential reason for this was thought to be related to epileptogenesis of KA-induced models^{10,11}. Several studies including the mechanisms of seizure generation and propagation have reported that strong c-Fos expression of DG was observed even when behavioral or electrographic seizures were suppressed¹²⁻¹⁴. In this context, we can consider that closed-loop SNF stimulation may not be able to prevent the biological phenomenon of potentially acting KA in DG, even if it had consistently suppressed seizure activities during acute experiments. Thus, it is considered that no reduction of c-Fos activation in DG after SNF stimulation was observed compared to other regions.

Low-intensity WF stimulation (LIWF; WF stimulation at its safety level) induced less suppressing effects than the SNF as confirmed by analyzing the brainwave power spectrum. However, LIWF still showed a significant decrease in the power spectrum compared to that of the non-stimulated group. Although LIWF stimulation due to its low intensity did not noticeably reduce c-Fos expression in the hippocampal region where KA was strongly activated, it is thought that stimulation energy delivered through the extrahippocampal connecting pathway may have prevented or suppressed neuronal cells from hyperactivation in the cortex. Furthermore, the reduction of c-Fos positive cells in the cortex after LIWF stimulation agrees well with the attenuation of the power spectrum during LIWF experiments compared to the non-stimulated condition in Figure 5.

To confirm the above statements, we believe that further studies on the cellular or network level mechanisms of SNF stimulation using brain slices should be conducted to confirm the exact rationale for these phenomena. We have added these considerations to the Discussion section of the revised manuscript.

Modifications:

Page 20, line 462–474: “Although the field distribution was restricted to the hippocampal region ... to confirm the exact rationale for these phenomena.”

- 3) It is stated that number of GAD65 neurons was quantified to “verify the contribution of GABAergic neurons to the seizure suppressing effects observed during stimulation.” Identifying just the number of GAD65 positive neurons does not allow interpretation about functional properties during a stimulation epoch.

Response: We appreciate the reviewer for this important comment regarding the interpretation of GAD65 neurons. As the reviewer pointed out, we agree that we overinterpreted the experimental results in the previous manuscript. Thus, we modified text in more appropriate manner in the Results section as below.

Modifications:

Page 14, line 300–302: “Second, to identify changes in neuronal inhibitory synaptic activity following stimulation, the molecular marker GAD65 was also used to analyze expression level changes in brain regions.”

5. Although including the chronic seizure model is laudable, again, the comparison here should be between WF and SNF stimulation rather than just SNF and non-stimulated animals. Furthermore, the behavioral tests are only performed during SNF stimulation; would WF stimulation, which causes off target effects, cause detrimental effects on these tasks? If not, the tasks may not be sensitive enough to capture such effects.

Response: We appreciate the reviewer for this valuable comment regarding the lack of comparison between WF and SNF conditions in chronic experiments and behavioral tests. Although our original plan was based on the assumption that demonstrating the superiority of SNF over WF and non-stimulation via the acute study and demonstrating the superiority of SNF over non-stimulation via the chronic study would be enough, we see that the addition of WF in chronic study would complete the flow much better. As recommended, we conducted additional experiments to compare the therapeutic effects of WF versus SNF stimulation using a chronic seizure model and determine detrimental effects due to the spreading fields during WF stimulation in an awake rodent model. These new experimental results using awake rats showed very similar results to those of the acute experiments and have been added to the Results section of the manuscript.

Modifications:

Page 15, line 334–348: “We first modeled chronic TLE rats with KA injections and implanted ... whereas significant differences were observed in the WF stimulation group (Fig. 8f, h, i).”

6. In the “Anti-seizure effects of hippocampal intervention” paragraph, it would be important to specify the mean and range of latencies at which seizures are terminated after stimulation.

Response: We appreciate the reviewer for this suggestion. As recommended, we added a plot presenting the distribution of latencies in Figure 2 and modified text to further explain the figure in the Results section of the revised manuscript as below.

Modifications:

Page 8, line 155–157: “Approximately 70% of seizures were terminated with hippocampal stimulation that intervened in ongoing ictal activities (Fig. 2g, h; termination latency after stimulation: 4.00 (IQR = 2.00–8.00) s; n = 204 trials from 6 animals).”

7. Figure 2 is focused on WF stimulation in the acute model, and the conclusion is that this stimulation is highly effective in suppressing seizures. Subsequently, this same WF stimulation is shown to have a seizure termination rate of only 0.57. The organization and presentation of the data make the manuscript extremely difficult to follow.

Response: We highly appreciate for this comment. We believe that this comment arose from the ambiguity of our writing. To validate the proposed SNF modality, we took phased approaches; in the first phase, we tried to validate the overall hippocampal stimulation for the epileptic control using WF stimulation and then demonstrate the superiority of SNF over WF in a later phase. During the phased approach, the WF stimulation in Fig. 2 had no intensity limit and the WF stimulation in Fig. 5 had a safety limit, thus a significant difference in seizure termination rates was observed between the two experiments. However, in reviewing this valuable comment, we realized that the different experiments in different phases mistakenly have the same group name, making it difficult for the reader to follow (e.g. WF stimulations in Fig. 3 and 5 had the same group name, but they actually used

different stimulation intensities). Therefore, throughout the manuscript, we modified the names of all groups to be clearly distinguishable for improving readability as below.

- 1) Wide-field stimulation (WF-stim; no safety limit): used in Fig. 3, 4, and 8e-i.
- 2) Low-intensity wide-field stimulation (LIWF-stim; WF-stim with a safety limit): used in Fig. 5, 6, 7, and 8c, d.
- 3) Sequential narrow-field stimulation (SNF-stim): used in Fig. 4, 5, 6, 7, and 8.
- 4) Non-stimulation (Non-stim; control group with KA injection and no treatment): used in Fig. 5, 6, 7, and 8.

8. In the introduction, please note that it should be “temporally” and not “temporarily.”

Response: We really appreciate the reviewer for the careful review of the manuscript. At this time, we carefully checked the grammar throughout the manuscript one more time and corrected all the typos and errors.

References

- 1 Hellier, J. L., Patrylo, P. R., Buckmaster, P. S. & Dudek, F. E. Recurrent spontaneous motor seizures after repeated low-dose systemic treatment with kainate: assessment of a rat model of temporal lobe epilepsy. *Epilepsy research* **31**, 73-84 (1998).
- 2 Williams, P. A. *et al.* Development of spontaneous recurrent seizures after kainate-induced status epilepticus. *Journal of Neuroscience* **29**, 2103-2112 (2009).
- 3 Rao, M. S., Hattiangady, B., Reddy, D. S. & Shetty, A. K. Hippocampal neurodegeneration, spontaneous seizures, and mossy fiber sprouting in the F344 rat model of temporal lobe epilepsy. *Journal of neuroscience research* **83**, 1088-1105 (2006).
- 4 Rao, M. S., Hattiangady, B., Rai, K. S. & Shetty, A. K. Strategies for promoting anti-seizure effects of hippocampal fetal cells grafted into the hippocampus of rats exhibiting chronic temporal lobe epilepsy. *Neurobiology of disease* **27**, 117-132 (2007).
- 5 Nissinen, J., Halonen, T., Koivisto, E. & Pitkänen, A. A new model of chronic temporal lobe epilepsy induced by electrical stimulation of the amygdala in rat. *Epilepsy research* **38**, 177-205 (2000).
- 6 Tsipouras, M. G. Spectral information of EEG signals with respect to epilepsy classification. *EURASIP Journal on Advances in Signal Processing* **2019**, 10 (2019).
- 7 Zheng, Y. *et al.* Acute seizure control efficacy of multi-site closed-loop stimulation in a temporal lobe seizure model. *IEEE Transactions on Neural Systems and Rehabilitation Engineering* **27**, 419-428 (2019).
- 8 Xu, K. *et al.* An energy efficient adaboost cascade method for long-term seizure detection in portable neurostimulators. *IEEE Transactions on Neural Systems and Rehabilitation Engineering* **27**, 2274-2283 (2019).
- 9 Armstrong, C., Krook-Magnuson, E., Oijala, M. & Soltesz, I. Closed-loop optogenetic intervention in mice. *Nature protocols* **8**, 1475-1493 (2013).
- 10 Popovici, T., Represa, A., Barbin, G., Beaudoin, M. & Ben-Ari, Y. Effects of kainic acid-induced seizures and ischemia on c-fos-like proteins in rat brain. *Brain research* **536**, 183-194 (1990).
- 11 Gall, C., Murray, K. & Isackson, P. J. Kainic acid-induced seizures stimulate increased expression of nerve growth factor mRNA in rat hippocampus. *Molecular brain research* **9**, 113-123 (1991).
- 12 Lily, M., Polygalov, D., Wintzer, M. E., Chiang, M.-C. & McHugh, T. J. CA3 synaptic silencing attenuates kainic acid-induced seizures and hippocampal network oscillations. *Eneuro* **3** (2016).
- 13 Lu, Y. *et al.* Optogenetic dissection of ictal propagation in the hippocampal–entorhinal cortex structures. *Nature communications* **7**, 1-12 (2016).
- 14 Kay, H. Y., Greene, D. L., Kang, S., Kosenko, A. & Hoshi, N. M-current preservation contributes to anticonvulsant effects of valproic acid. *The Journal of clinical investigation* **125**, 3904-3914 (2015).

Reviewers' comments:

Reviewer #2 (Remarks to the Author):

The authors gave clear answers to my question and made important improvement of the manuscript. I have no more question and would like to recommend acceptance of this paper.

Reviewer #3 (Remarks to the Author):

The authors have performed additional experiments and analysis, increasing the clarity of the results. The WF behavioral experiments were particularly helpful. Questions regarding the mechanistic effects of the stimulation on hippocampal, cortical, and white matter structures remain, but are conceivably beyond the scope of this work. I do not have additional concerns regarding the methods. Conclusions have been tempered appropriately in the Discussion.

Reviewers' comments

Reviewer 2:

1. The authors gave clear answers to my question and made important improvement of the manuscript. I have no more question and would like to recommend acceptance of this paper.

Response: We highly appreciate the reviewer for the feedbacks which were extremely helpful in improving the manuscript. Responding to the reviewer's comments, substantial improvements could be made via additional analyses. In addition, multiple discussion points were made clearer.

Reviewer 3:

1. The authors have performed additional experiments and analysis, increasing the clarity of the results. The WF behavioral experiments were particularly helpful. Questions regarding the mechanistic effects of the stimulation on hippocampal, cortical, and white matter structures remain, but are conceivably beyond the scope of this work. I do not have additional concerns regarding the methods. Conclusions have been tempered appropriately in the Discussion.

Response: We appreciate the reviewer for the positive feedbacks on the improvements made in the revised manuscript with additional experiments and analyses, and increased clarity on the results. More importantly, we completely agreed with the reviewer that further mechanistic investigation is needed to confirm the exact mechanism and phenomena of the therapeutic effects during SNF stimulation. This work will be immediately followed.